# Phenotype of circulating tumor-reactive T cells predicts immune checkpoint inhibitor response in non-small cell lung cancer

Katsuhiro Ito [1,2], Kei Iida [3,4], Tomoko Hirano [1,5], Merrin Man Long Leong [1], Kenji Morii[1], Toshi Menju[6], Hiroshi Date[6], Hiroaki Ozasa[7], Hironori Yoshida[7], Toyohiro Hirai[7], Shusuke Kawashima[8], Kazuhiro Aoyama [8], Yuka Saeki[8], Takashi Inozume[8], Takashi Kobayashi [2], Kenji Chamoto [1,9] ✉ & Tomonori Yaguchi [1,9,10] ✉

Peripheral blood (PB) is a source of tumor-infiltrating tumor-reactive T cells (TR-T). Circulating TR-Ts (cTR-T) in PB are expected to contribute to the efficacy of immune checkpoint inhibitors (ICIs), but their phenotype remains poorly understood. Here we analyse paired tumor-infiltrating and peripheral CD8⁺ T cells from patients with non-small cell lung carcinoma (NSCLC), using single-cell RNA and T cell receptor (TCR) sequencing. Tumor-infiltrating TR-Ts are defined based on the reported TR-T-associated gene signatures. Using their TCR sequence as a barcode, we identify cTR-Ts and their specific surface markers, including CD49a, CD49b, and HLA-DR. Trajectory analysis assigns a progenitor-like phenotype to cTR-Ts, suggesting a potential developmental relationship with tumor-infiltrating TR-Ts. By single-cell transcriptomic and flow cytometric analysis on an ICI-treated cohort we show that pre-treatment cTR-Ts in responders are characterized by a relatively low expression of exhaustion-related CD38. Following the first dose, cTR-Ts of responders transit towards a *TCF7*⁺ stem-like phenotype. Additionally, we validate cTR-T's phenotypic changes following PD-1 blockade therapy in mouse tumor models with artificial antigen. These findings suggest that the phenotypic state and transition of cTR-Ts may reflect their functional potential after tumor infiltration and are associated with therapeutic outcomes of ICIs.

Current successes in cancer immunotherapies, such as immune checkpoint inhibitors (ICI) and adoptive T cell therapies, highlight the critical role of T cell immunity for controlling solid cancers[1,2]. Tumor cells express antigens that distinguish them from normal cells. T cells recognize and eliminate tumors with antigen specificity. However, even within tumor-infiltrating lymphocytes (TILs), most T cells are bystander T cells that are tumor-non-specific, such as viral-specific T cells[3]. Therefore, it is crucial to understand the phenotypes and

[1]Department of Immunology and Genomic Medicine, Center for Cancer Immunotherapy and Immunobiology, Graduate School of Medicine, Kyoto University, Kyoto, Japan. [2]Department of Urology, Graduate School of Medicine, Kyoto University, Kyoto, Japan. [3]Infomatics Platform, Center for Cancer Immunotherapy and Immunobiology, Graduate School of Medicine, Kyoto University, Kyoto, Japan. [4]Faculty of Science and Engineering, Kindai University, Osaka, Japan. [5]Department of Dermatology, Graduate School of Medicine, Kyoto University, Kyoto, Japan. [6]Department of Thoracic Surgery, Graduate School of Medicine, Kyoto University, Kyoto, Japan. [7]Department of Respiratory Medicine, Graduate School of Medicine, Kyoto University, Kyoto, Japan. [8]Department of Dermatology, Chiba University Graduate School of Medicine, Chiba, Japan. [9]Department of Immuno-oncology PDT, Graduate School of Medicine, Kyoto University, Kyoto, Japan. [10]Department of Immune Metabolism, Center for Cancer Immunotherapy and Immunobiology, Graduate School of Medicine, Kyoto University, Kyoto, Japan. ✉e-mail: chamoto.kenji.4w@kyoto-u.ac.jp; yaguchi.tomonori.4m@kyoto-u.ac.jp

dynamics of tumor-reactive T cells (TR-T) rather than those of bulk T cells when studying tumor immunity.

PD-1 blockade therapies, a major class of ICIs, function by disrupting the PD-1 co-inhibitory pathway to activate TR-Ts. Initially, ICI were believed to primarily reinvigorate exhausted or dysfunctional CD8+ TILs. This hypothesis has led to extensive research comparing TIL phenotypes between responders and non-responders[4–7]. Accumulating evidence now indicates that terminally exhausted T cells within tumor tissues are largely refractory to PD-1 blockade[5,8]. Two alternative sources of functional effector T cells capable of exerting cytotoxic antitumor activity have been proposed: a small subset of early dysfunctional TR-Ts within the tumor or TR-Ts recruited from outside the tumor, such as peripheral blood or tumor-draining lymph nodes[9]. Supporting the latter hypothesis, several studies have reported clonal replacement of intratumoral T cells following PD-1 blockade[10,11]. In addition, early studies have shown that increased proliferation of circulating lymphocytes predicts better responses to ICI[12,13], and that clonotypic expansion of peripheral T cells correlates with clinical benefit[14]. However, tumor reactivity was not addressed in most of these studies. As a result, the differentiation trajectory of circulating TR-Ts from the peripheral blood to the tumor, as well as the influence of ICI on this process, remains incompletely understood. A major obstacle is the rarity of circulating TR-Ts in peripheral blood, which is typically less than 0.1%[15–18], and the lack of established specific markers to identify them.

Identifying TR-Ts requires assessing their reactivity to autologous tumor cells or tumor antigens. However, acquiring sufficient tumor material or identifying relevant antigens is often labor-intensive and technically challenging. To overcome these limitations, numerous studies have sought to define surrogate markers for TR-Ts. Candidate markers identified in tumor-infiltrating TR-Ts include immune checkpoint molecules such as PD-1, CD39, or TIM3[3,19–23], tissue residency-related molecules such as CD103[19,24], and CXCL13[25,26]. Because the environment surrounding TR-Ts differs between tumors and peripheries, peripheral TR-Ts are expected to have a distinct phenotype. However, current studies on peripheral TR-Ts are limited and often rely on known tumor antigens, such as MART1 (melanoma antigen recognized by T cells 1), or those predicted through next-generation sequencing[15,16,18,24]. As a result, they have captured only a limited portion of the circulating TR-Ts population. Furthermore, the relationship between the phenotype of peripheral TR-Ts and the efficacy of PD-1 blockade therapy has not yet been investigated.

In this study, we aim to characterize circulating TR-Ts in the peripheral blood and elucidate their contribution to antitumor immunity during PD-1 blockade therapy. We use paired single-cell sequencing of peripheral blood mononuclear cells (PBMC) and TILs from 9 patients with non-small cell lung carcinoma (NSCLC) to assess the phenotype and differentiation trajectories of TR-Ts in both PBMCs and TILs. Tumor-infiltrating TR-Ts are defined based on previously reported TR-T-associated gene signatures. Using the T-cell receptor (TCR) as a clonal barcode, we successfully identify circulating TR-Ts that shared identical TCRs with tumor-infiltrating TR-Ts. These peripheral TR-Ts specifically express HLA-DR, CD49a, and CD49b, the latter two of which are known as integrins necessary for tissue retention. Furthermore, we find that both the pre-treatment phenotypes and post-treatment cell fate of peripheral TR-Ts differ between responders and non-responders to PD-1 blockade therapy. These findings provide insight into the dynamic relationship between peripheral and tumor TR-Ts and how their differentiation states may influence therapeutic response.

## Results
### Characterization of predicted TR-T and bystander T cells in tumors
We generated a single-cell RNA-seq/TCR-seq/CITE-seq dataset of CD8+ T cells isolated from resected tumor tissues and peripheral blood of

nine patients with NSCLC without EGFR mutations (Fig. 1A and Supplementary Table 1). First, we characterized the transcriptional features of TIL. 29,509 CD8+ TIL passed quality control and were integrated to obtain broad CD8+ TIL subsets, as visualized using uniform manifold approximation and projection (UMAP) dimensionality reduction (Fig. 1B and Supplementary Fig. 1A, B). The expression of subset-defining markers and T cell checkpoint molecules was visualized using a heatmap (Fig. 1C). We labeled CD8+ TILs based on the previous studies to profile TR-Ts in NSCLC and other cancers[27,28], and CD8+ TIL were clustered based on exhaustion, memory, tissue residence, and effector function programs. This clustering pattern is consistent with that of previous studies[29,30], indicating that our samples represent a heterogeneous population of CD8+ TILs from NSCLC.

Early studies have profiled the phenotype of TR-Ts in TILs based on experimentally confirmed antigen specificity[27,28]. In the present study, rather than focusing on validating antigen specificity, we aimed to define TR-T in a more comprehensive manner by applying a gene signature-based approach. We applied the MANAscore reported by Zeng et al.[31]. This score consists of two positive genes (CXCL13 and ENTPD1) and one negative gene (IL7R) and can detect TR-Ts with higher sensitivity than other RNAseq-based gene signatures. The MANAscore was enriched in TRM(2)/Exhausted clusters of CD8+ TILs (Fig. 1D), which is consistent with earlier findings showing that TR-Ts exhibit a tissue-resident and exhausted phenotype[27,28].

The expression of other genes and proteins known to predict tumor reactivity, such as CD39[3,19,20,26], CXCL13[25,26], CD103[19,24], and another TR-T prediction signature score reported by Lowery et al.[23] or Hanada et al.[26] were enriched in the same clusters (Supplementary Fig. 1C and Supplementary Data 1). Using the public complementary determinant region 3 (CDR3) database[32], we identified 71 clonotypes with CDR3 reactivity to common viral antigens, including influenza, cytomegalovirus, and Epstein-Barr virus (Fig. 1E). Most of these virus-reactive T cells (VR-Ts) are located in effector or TRM[1] clusters, which aligns with previous findings that bystander T cells are not exhausted[3,28]. To define TR-Ts, We first select clonotype in which at least ≥5% of cells were MANAscore-positive. From the T cell clones which meet the criteria, we further filtered out small (<1%) clones because intratumoral clonal expansion is a critical feature of TR-Ts[19,20], resulting in 93 predicted TR-T (pTR-T) clones, which accounted for 10839 cells in the total CD8+ T cell population (Fig. 1F). Although pTR-Ts were mainly distributed in exhausted and tissue-resident clusters, some cells of these clones exhibited memory and effector phenotypes. The distribution pattern of pTR-Ts varied among patients (Supplementary Fig. 1D).

Next, we examined the correlation between the proportion of pTR-Ts in TILs and the total number of tumor-infiltrating CD8+ T cells. The proportion of pTR-Ts among CD8+ TIL varied among patients (0–67.2%) and showed a positive correlation with the total number of the tumor-infiltrating CD8+ T cells, as assessed in the formalin-fixed paraffin-embedded (FFPE) pathology specimens (Supplementary Fig. 2A). In contrast, the proportion of VR-Ts did not correlate (Supplementary Fig. 2B). Collectively, we defined pTR-Ts in our NSCLC TIL datasets based on TR-T-associated gene signature and their clonal size.

### Circulating pTR-T expresses unique markers of activation and tissue residency
Next, we detected pTR-T clones in PBMC. To this end, we collected 46,572 peripheral CD8+ T cells from PBMCs of the same cohort and performed single-cell RNA-seq using the same procedure as for TILs (Fig. 1A). Unsupervised clustering of CD8+ T cells in PBMCs identified 15 distinct clusters. When mapped onto established T cell subsets based on known gene and surface protein markers, these clusters corresponded to naïve (cluster 5, 7), central memory (cluster 6), effector memory (cluster 3, 8, 11), and terminally differentiated effector memory T cells re-expressing CD45RA (TEMRA) (cluster 0, 1, 2, 4, 10, 12)

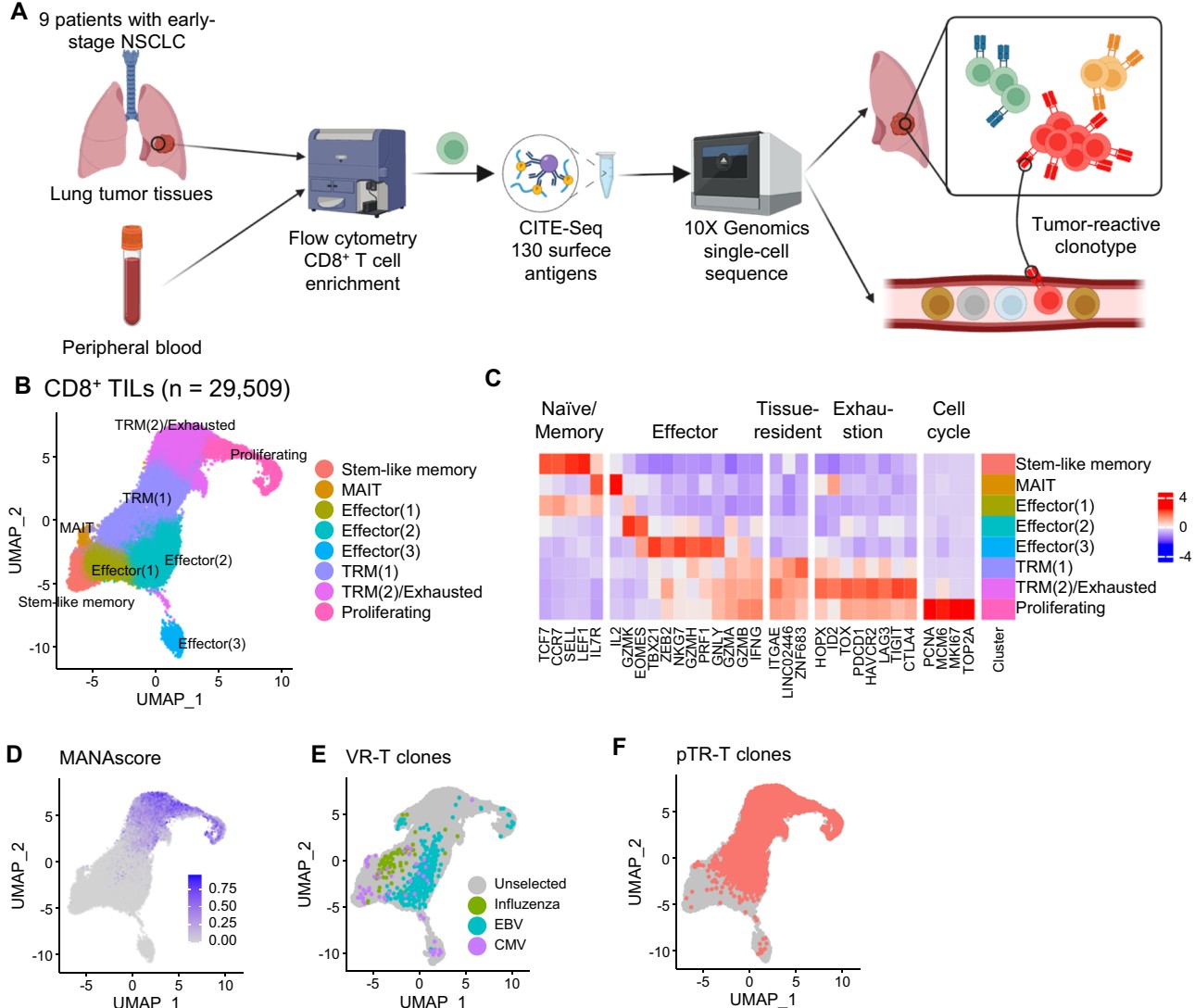

**Fig. 1 | Detection of TR-Ts from high-resolution single-cell analysis. A** Paired TILs and PBMCs were collected from 9 patients and underwent single-cell RNA/TCR/CITE-seq. Created in BioRender. Ito, K. (2025) https://BioRender.com/g10fuad. **B** High-resolution single-cell CD8+ TILs mapped to 8 clusters. **C** Heatmap of subset-defining gene signatures. **D** Neoantigen-reactive score (MANAscore) calculated and shown in UMAP. **E** Distribution of VR-Ts on UMAP. VR-Ts were defined as clones containing complementary determining region 3 sequences reactive to influenza, CMV, or EBV (**F**) pTR-T clones mapped in UMAP. Clones with MANAscore+ cells ≥ 5% and frequency ≥ 1% among TILs were predicted as TR-Ts. *TR-T* tumor-reactive T cells; *TIL* tumor-infiltrating lymphocytes; *PBMC* peripheral blood mononuclear cells; *UMAP* uniform manifold approximation and projection; *VR-T* viral-reactive T cells; *CMV* cytomegalovirus; *EBV* Epstein Barr virus.

(Fig. 2A and Supplementary Fig. 3A). Mucosal-associated invariant T cells (cluster 9) and proliferative cells (cluster 14) are localized to distinct and distant regions on the UMAP. Using the TCR sequence information of pTR-Ts in TIL as a hashtag, we detected 26 T-cell clones comprising 159 of the 46,572 peripheral CD8+ T cells. These were defined as circulating pTR-T clones (Fig. 2B). Circulating pTR-Ts were detected in only six patients (TS04, 13, 16, 19, 20, and 25) with varied frequency (0.2–1.1%) (Supplementary Fig. 3B). The frequency of pTR-Ts in PBMC did not correlate with its frequency in TILs at the patient level (Supplementary Fig. 3C). However, when comparing the frequency of each clone in the peripheral blood and tumor tissue, pTR-Ts were significantly enriched in tumors, while VR-Ts did not exhibit a similar expansion (Supplementary Fig. 3D). This finding suggests that TR-Ts preferentially accumulate in the tumor microenvironment, whereas VR-Ts do not exhibit a specific localization pattern.

We evaluated the cluster in which the circulating pTR-Ts were preferentially detected and found that they were enriched in the effector memory (2, 3, 8 11) and proliferating cluster[14], particularly in cluster 11 (Fig. 2B, C and Supplementary Fig. 3E), suggesting that they might have a unique gene expression pattern. This previously uncharacterized subset of effector memory T cells (cluster 11) had unique transcriptional programs including high expression of activation-related markers such as *CD69*, *CD74*, and *HLA-DRA*, whereas innate immunity-related genes such as *GNLY*, *GZMB* and *FGFBP2* were downregulated (Supplementary Fig. 3A and Supplementary Data 2). This cluster also expresses GZMK, which is emerging as a key molecule related to tumor reactivity and ICI response (Supplementary Fig. 3A)[11,33,34]. The enrichment of pTR-Ts in the GZMK+ cluster suggests that these cells may contribute to antitumor immunity in the peripheral blood. In contrast, the circulating VR-Ts were distributed throughout the clusters (Supplementary Fig. 3F). Differentially expressed gene (DEG) and protein (DEP) analyses between circulating pTR-Ts and other peripheral CD8+ T cells revealed the unique properties of pTR-Ts in PBMCs (Fig. 2D, E). In detail, pTR-Ts are characterized by high expression of MHC class II-related genes (*HLA-DR* and *CD74*), chemokines (*CXCR3* and *CCL5*), and tissue residency-

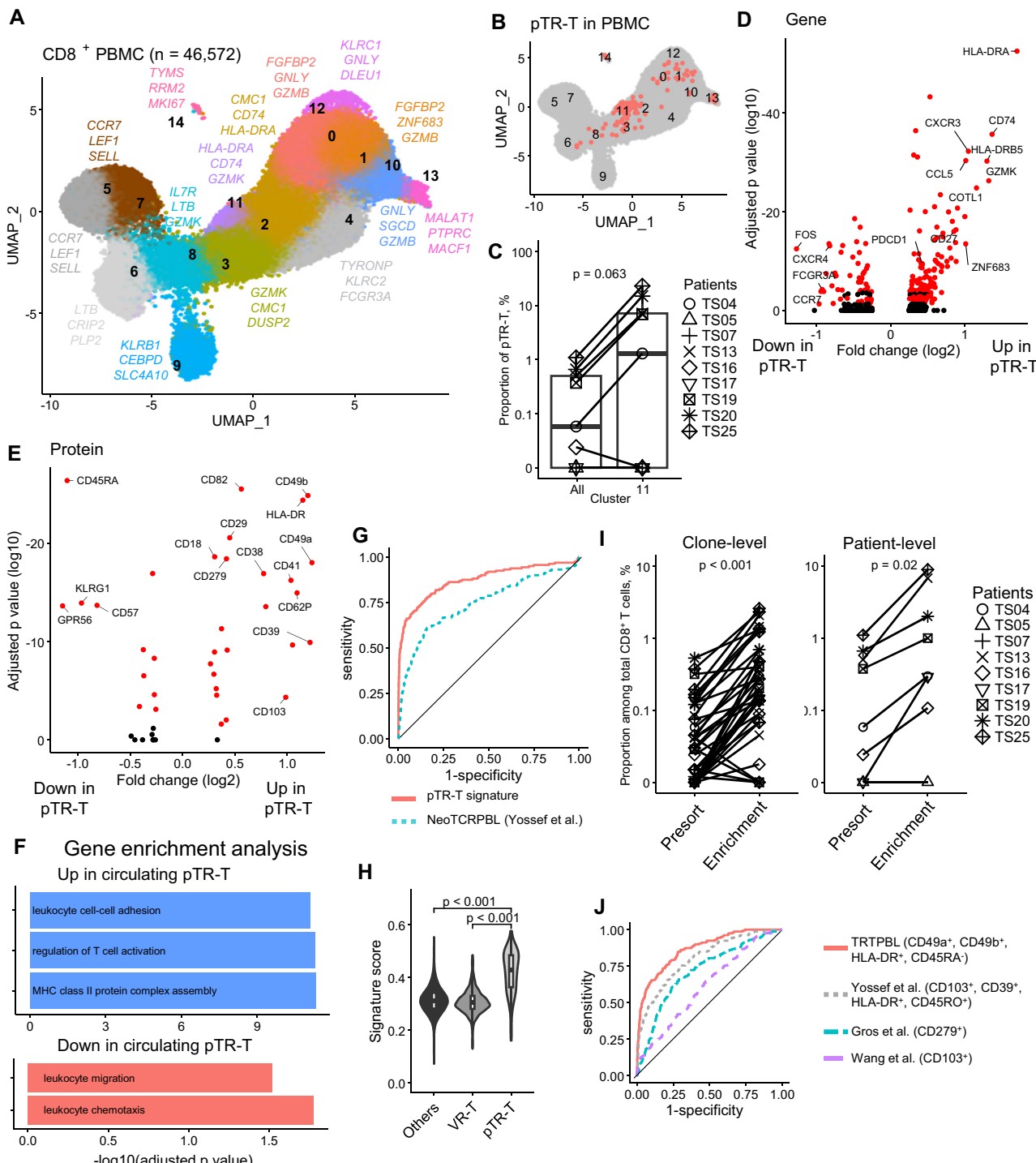

**Fig. 2 | TR-Ts in PBMC express unique markers. A** Single-cell CD8[+] PBMCs from 9 patients were mapped to 15 clusters (**B**) pTR-T clones defined in Fig. 1 were mapped to the UMAP of CD8[+] PBMCs .**C** Proportion of pTR-T in cluster 11 and all CD8[+] T cells stratified by patients ($n = 9$). In the box plots, the center line indicates the median; the box represents the interquartile range (25th–75th percentiles); whiskers extend to the most extreme values within 1.5 × the interquartile range. (two-sided paired Wilcoxon rank-sum test). **D** DEGs between circulating pTR-Ts and other CD8[+] T cells. Genes that were consistently up- or downregulated across patients (consistency score > 0.33, as shown in Supplementary Fig. 4A) are highlighted in red. *P*-values were adjusted for multiple comparisons using the Bonferroni correction. **E** DEPs between circulating pTR-Ts and other CD8[+] T cells. Proteins that were consistently up- or downregulated across patients (consistency score > 0.33, as shown in Supplementary Fig. 4B) are highlighted in red. *P*-values were adjusted for multiple comparisons using the Bonferroni correction. **F** GO enrichment analysis of upregulated and downregulated genes in pTR-Ts. *P*-values were adjusted for

multiple comparisons using the Benjamini–Hochberg method. **G** ROC analysis for detecting circulating pTR-Ts based on gene signature score derived from the pTR-T gene signature (Supplementary Data 1) and NeoTCRPBL signature reported by Yossef et al. **H** Comparison of the pTR-T gene signature score among pTR-Ts ($n = 159$), VR-Ts ($n = 1467$), and other cells ($n = 44946$). In the box plots, the center line indicates the median; the box represents the interquartile range (25th–75th percentiles); whiskers extend to the most extreme values within 1.5 × the interquartile range. (two-sided unpaired *t* test). **I** Flow-cytometry enrichment of pTR-Ts based on TRTPBL markers at clone-level (left, $n = 41$) and patient-level (right, $n = 8$). (two-sided paired Wilcoxon rank-sum test). **J** ROC analysis of surface protein markers for detecting circulating pTR-Ts, including TRTPBL marker and previously reported markers. *DEG* differentially expressed genes; *GO* gene ontology; *DEP* differentially expressed proteins; *ROC* receiver operating characteristic; *TRTPBL* protein-based pTR-T marker.

related genes (*ZNF683*). They downregulated both naïve (*CXCR4*, *CCR7*, and *FOS*) and terminal effector (*FCGR3A* and *KLRG1*) genes (Supplementary Data 3). DEP analysis revealed the expression of integrin-related proteins such as CD49a (*ITGA1*), CD49b (*ITGA2*), and CD103 (*ITGAE*) increased in pTR-Ts, as well as HLA-DR (Supplementary Data 4). *ITGA1*, *ITGA2* and *ITGAE* were not detected in transcript-level differential expression analysis, indicating the significance of post-transcriptional modifications. CD279 (*PDCD1*), a conventional marker of neoantigen-specific T cells in PBMC[15,35,36], was identified as a DEG and DEP; however, the magnitude of the difference was small. Following the cell-level DEG and DEP analyses, we performed DEG and DEP analyses across individual patients to minimize patient-specific effects and used DEGs that were consistently observed in multiple patients for subsequent analyses (Fig. 2D, E, Supplementary Fig. 4A, B and Supplementary Data 5 and 6). Gene ontology enrichment analysis showed that the upregulated genes in pTR-Ts were related to Leucocyte Cell-Cell Adhesion (GO:0007159), Regulation of T cell Activation (GO:0050863), and MHC Class II Protein Complex Assembly (GO:0002399). Downregulated genes in pTR-Ts include Leucocyte Migration (GO:0050900) and Leucocyte Chemotaxis (GO:0030595) (Fig. 2F). These data suggest that circulating pTR-Ts are activated, pre-resident state while reducing migratory potential.

Next, we aimed to create a transcriptomic signature to distinguish circulating pTR-Ts from other circulating CD8+ T cells. Although a circulating TR-T gene signature (NeoTCRPBL) has been reported[16], it was derived from melanoma, colorectal, and breast cancers, and no circulating TR-T signature has been established for lung cancer. Therefore, we developed a lung cancer–derived circulating pTR-T signature to evaluate whether the transcriptional features of circulating TR-T cells in NSCLC are comparable to those reported in other tumor types. First, we calculated a gene signature score by evaluating the expression levels of 140 DEGs consistently upregulated in pTR-Ts across patients (Fig. 2D, Supplementary Fig. 4A and Supplementary Data 1 and 5, see "Method"), which we named the "**pTR-T gene signature score**." We compared our circulating pTR-T signature with the NeoTCRPBL signature and found that only ~ 30% of genes overlapped (Supplementary Fig. 4C). Our signature showed a higher area under the curve (AUC = 0.87) for identifying pTR-Ts in receiver operating characteristic (ROC) analysis compared with NeoTCRPBL (AUC = 0.75) (Fig. 2G), indicating improved performance in predicting peripheral TR-Ts. These results suggest that the characteristics of pTR-Ts may be affected by tumor type. The sensitivity was 78.0% and the specificity was 83.9% when the optimal cut-off was determined by the Youden index. Circulating pTR-Ts showed a higher signature score than VR-Ts or other circulating T-cells (Fig. 2H).

For clinical applications, we developed a surface protein-based marker combination to detect circulating TR-Ts. DEPs identified in this study and consistently observed across patients were selected as candidate markers (Fig. 2E and Supplementary Fig. 4B). We observed that a combination of the top three positive markers (CD49a, CD49b, and HLADR) and negative markers (CD45RA, GPR56 and KLRG1) could effectively distinguish pTR-Ts (Supplementary Fig. 4D). The performance remained largely unchanged when either negative marker was excluded. Given its widespread use in clinical settings, we selected CD45RA as a representative negative marker. CD49a and CD49b showed the highest fold changes. Approximately 70% of circulating pTR-T cells expressed CD49a, CD49b, or both. (Supplementary Fig. 4E). Biologically, CD49a and CD49b may be competitively expressed because both form a heterodimer with CD29[37]. Therefore, in the development of the marker panel, we adopted the criterion of (CD49a+ or CD49b+), meaning that cells expressing either CD49a or CD49b were considered positive. Finally, we defined the protein-based peripheral blood (PBL) pTR-T marker, named "**TRTPBL marker**," as CD45RA-, HLADR+, and (CD49a+ or CD49b+). We sorted TRTPBL marker–positive cells and performed single-cell TCR sequencing. The sorted population showed a significant enrichment of pTR-Ts at both the clonal and patient levels (Fig. 2I). The concentration of pTR-Ts increased 2.7-13.5 folds. Notably, sorting based on TRTPBL markers successfully identified 15 additional pTR-T clones that were not detected in the total CD8+ PBMCs scRNA dataset. Previous literature reported CD279, CD103, or combination of CD103, HLA-DR, CD39, and CD45RO as surface-protein based markers of circulating TR-Ts[15,16,24]. Our TRTPBL marker showed the highest discriminative ability to detect circulating pTR-T compared with previously reported markers (Fig. 2J).

To validate our "**pTR-T gene signature score**" and "**TRTPBL marker**," we investigated whether this score and marker can successfully identify predefined true TR-Ts. We collected CD8+ TILs and PBMCs from a patient with melanoma, where T cells specific to MART1, a known melanoma antigen, could be detected with MART1/HLA0201 tetramer (Supplementary Fig. 5A). Particularly, the fresh CD8+ TILs from this patient with melanoma contained MART1-specific T cells at a rate of 1.0% as determined by the MART1/HLA0201 tetramer assay (Supplementary Fig. 5B). Single-cell TCRseq of these MART1/HLA0201 tetramer-positive T cells in TILs revealed that a single clonotype (clone 1) predominantly occupied >95% of total clones (Supplementary Fig. 5C and Supplementary Table 2). In parallel, a MART1-specific T cell line with the identical TCR (clone 1) was established from the cultured TILs in the presence of the MART1 peptide[38], and their functionality and specificity were validated using the IFN-γ releasing assay (Supplementary Fig. 5D). We conducted single-cell RNA/TCR/CITEseq of 11,895 peripheral CD8+ T cells and used query mapping to position these cells in a previously generated UMAP image (Fig. 2A). We detected three MART1-specific T cells (clone 1). The clone 1 were observed in cluster 3, and 14 (Supplementary Fig. 5E). The pTR-T gene signature score was significantly higher in circulating MART1-specific T cells than other peripheral CD8+ T cells (Supplementary Fig. 5F). The circulating MART1-specific T cells were CD45RA-negative, HLADR-positive, and CD49b-positive (Supplementary Fig. 5G). CD49a protein expression was moderate in MART1-specific T cells, indicating different expression patterns of integrins on circulating pTR-Ts, depending on the cancer type or patient. Finally, we compared the discriminative performance of our pTR-T gene signature and TRTPBL marker with previously reported markers (Supplementary Fig. 5H). Our TRTPBL marker showed a high AUC value (0.8235) to detect MART1-specific T cells from total CD8+ T cells in PBMC. These results suggest that circulating TR-T uniquely expresses our **TRTPBL marker** and could be enriched by these marker combinations in peripheral CD8+ T cells.

## Circulating pTR-T showed a progenitor phenotype of tumoral pTR-T

To better understand the relationship between circulating pTR-Ts and those residing in tumors, we examined their phenotypic and developmental characteristics. We investigated whether circulating pTR-Ts resemble specific differentiation states of tumoral pTR-Ts. To test this, we focused on pTR-T clones detected in both TILs and PBMC, generated a new UMAP in which TIL- and PBMC-derived pTR-Ts were plotted, and performed a pseudotime-trajectory analysis. PBMC-derived pTR-Ts were differentially mapped on the UMAP and located at the root of the trajectory (Fig. 3A). The root of the trajectory was characterized by high expression of *TCF7* and *IL7R* and intermediate expression of *PDCD1*, *HAVCR2*, *TIGIT*, and *CXCL13*, indicating the progenitor properties of circulating pTR-Ts (Fig. 3B). This finding was consistent across clones, as the exhaustion signature increased from PBMC to TIL, whereas the progenitor signature decreased (Fig. 3C). In contrast, the expression of TRTPBL markers (CD49a, CD49b, and HLA-DR) was relatively inconsistent across clones (Fig. 3D). Changes in the marker expression exhibited distinct patterns depending on the clone, even within the same patient (Supplementary Fig. 6). This heterogeneity may reflect differences in TCR affinity or in the stage of T-cell differentiation between peripheral blood and tumor-infiltrating

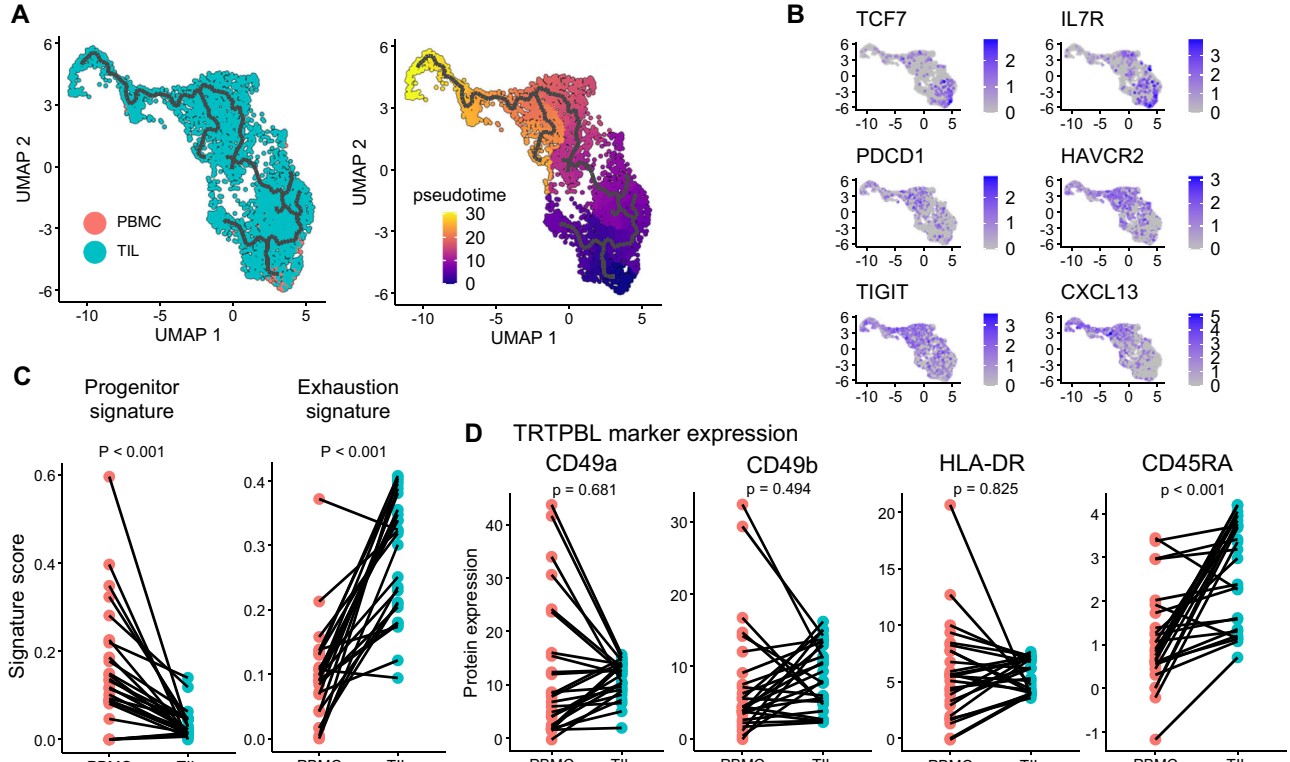

**Fig. 3 | Circulating TR-T is a progenitor of tumoral TR-T. A** pTR-T clones detected in TILs and PBMCs were integrated and projected onto a new UMAP. Trajectory analysis showed that circulating pTR-Ts were located at the root of the differentiation path (left). Pseudotime analysis indicated differentiation from circulating pTR-Ts toward those in TILs (right). **B** Representative exhaustion- and progenitor-related gene expression shown on the UMAP. **C** Averaged clone-level exhaustion and progenitor signature scores between pTR-Ts in PBMCs and those in TILs. Each dot and line represents a clonotype ($n = 26$). (two-sided paired $t$ test). **D** Averaged clone-level protein expression of TRTPBL marker (CD49a, CD49b, HLA-DR, CD45RA) between pTR-Ts in PBMC and those in TILs. Each dot and line represents a clonotype ($n = 26$). (two-sided paired $t$ test).

populations. Notably, despite these variable changes, TRTPBL markers were consistently and relatively highly expressed in both peripheral blood and TIL, suggesting that circulating pTR-Ts may already exhibit features of a tissue-resident (CD49a and CD49b) and activated (HLA-DR) phenotype. Collectively, our trajectory analysis demonstrated phenotypic and transcriptional distinctions and similarity between circulating and intratumoral TR-Ts.

### Circulating pTR-T changes their phenotype in ICI responders

Next, using our circulating pTR-T gene signature, we aimed to assess how the proportion and phenotype of circulating pTR-Ts differed between ICI responders and non-responders. We collected pre- and post-treatment PBMCs from four responders and four non-responders with NSCLC who received combination therapy of PD-1 blockade and chemotherapy and performed single-cell RNAseq for CD8+ T cells (Fig. 4A and Supplementary Table 3). To minimize patient-level bias, we selected patients with a history of smoking and with lung adenocarcinoma without EGFR mutations. We collected 171,373 peripheral CD8+ T cells (54,494 in pre-treatment and 116,879 in post-treatment) from eight patients and mapped the cells onto previous UMAP images (Fig. 2A) using query mapping (Supplementary Fig. 7A, B). First, we analyzed the pre-treatment samples. To investigate clone-level differences between responders and non-responders, we averaged the pTR-T gene signature score across each clone, and clones exceeding the cut-off were defined as pTR-Ts. The pTR-Ts in the pre-treatment samples (pre-treatment pTR-T) accumulate in cluster 11 in the UMAP, similar to the training data in Fig. 2B, regardless of the response to the treatment (Fig. 4B). Although the overall UMAP distribution was similar between responders and non-responders, we identified several DEGs and DEPs when comparing gene expression profiles between the two groups

(Figs. 4C, D and Supplementary Data 7 and 8). In addition, patient-level evaluation of these DEGs (Supplementary Fig. 8A) revealed consistently downregulated genes in non-responders, including *GRHPR* (glyoxylate reductase/hydroxypyruvate reductase), *MGMT* (methylguanine DNA methyltransferase), and *AKR7A2* (aldo-keto reductase). Although the roles of these genes in T cells remain largely unknown, their involvement in oxidative stress and DNA damage response suggests that metabolic adaptation of circulating TR-Ts may differ between responders and non-responders. Patient-level DEP analysis further revealed that HLA-DR and CD38 were consistently upregulated in non-responders (Supplementary Fig. 8B). CD38, a cell surface glycoprotein involved in immune regulation and NAD+ metabolism, has been reported to contribute to resistance to PD-1 blockade therapies[39–43]. Therefore, assessing these markers in pTR-Ts could be used for ICI response prediction. These phenotypic differences may further reveal underlying mechanisms that contribute to clinical outcomes.

We further assessed changes in the phenotypes of circulating pTR-Ts after ICI treatment. On the same UMAP as in Fig. 4B, we plotted the post-treatment pTR-Ts, defined as CD8+ T cells from post-treatment samples (after the first dose of the drug) that share the same TCR as pre-treatment pTR-T (Fig. 4E and Supplementary Fig. 8C, D). In responders, pTR-Ts in clusters 2 and 11 were significantly reduced after treatment, accompanied by a marked increase in clusters 3 and 8. In contrast, such shifts were not observed in non-responders (Fig. 4F). Cluster 3 and 8 represents a previously uncharacterized subset of effector memory T cells, co-expressing both effector-related genes (e.g., *CD28*, *GZMK*) and stemness-associated genes (e.g., *IL7R*, *TCF7*) (Supplementary Fig. 3A). This shift in responders suggests that pTR-Ts in responders acquired properties associated with long-term persistence. As a result, the frequency of TRTPBL

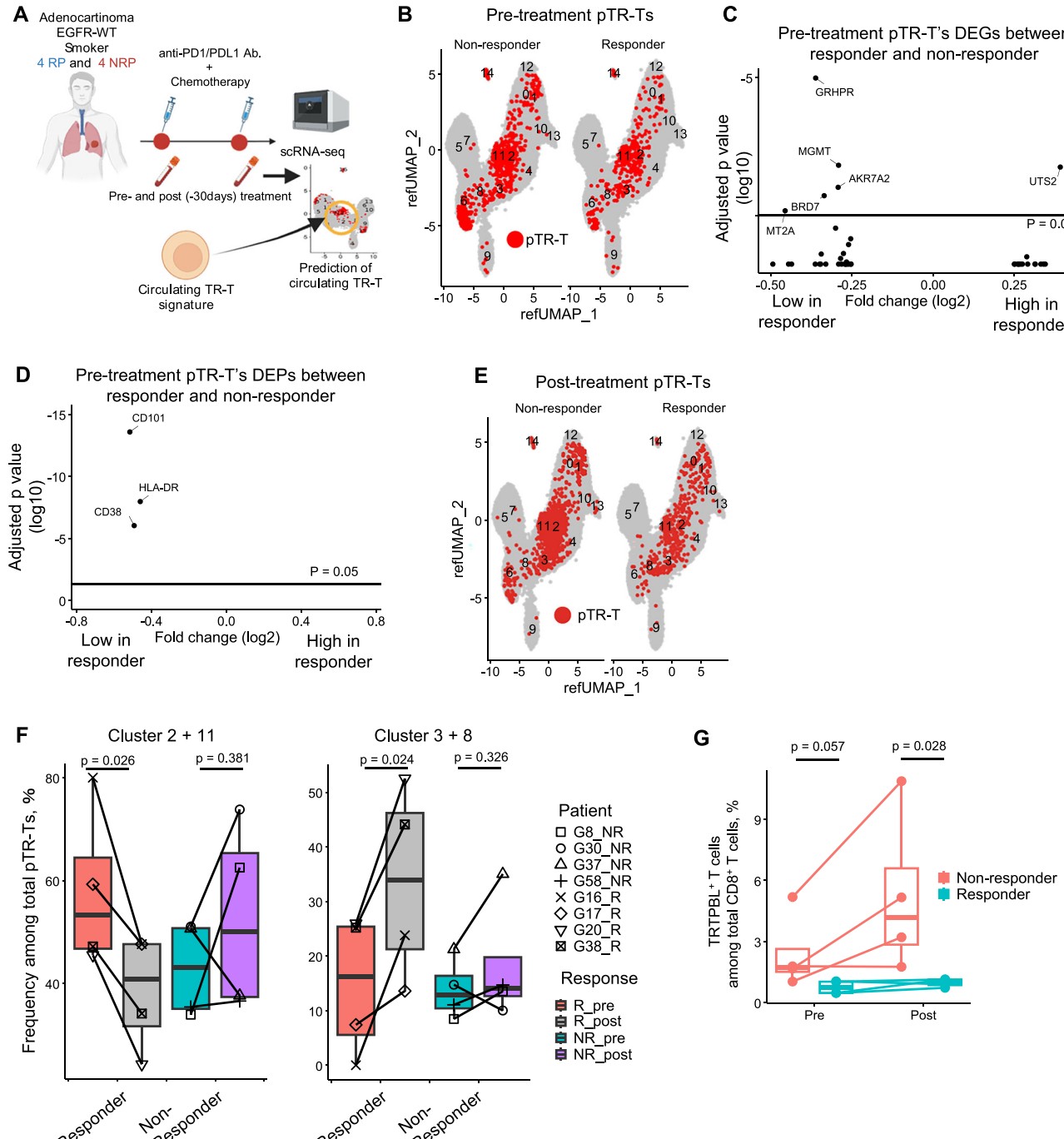

**Fig. 4 | Circulating pTR-T changes their phenotype in ICI responders. A** Sample collection scheme. Pre- and post-treatment CD8+ PBMCs from 4 responders and 4 non-responders to PD-1 blockade therapy underwent single-cell RNA/TCR/CITE-seq. Circulating pTR-Ts were predicted using the signature defined in Fig. 2. Created in BioRender. Ito, K. (2025) https://BioRender.com/2e2or46 **B** Distribution of pTR-Ts on UMAP. pTR-Ts are predominantly localized to cluster 11. **C** Volcano plots showing DEGs between responders' and non-responders' pTR-Ts. **D** Volcano plots showing DEPs between responders' and non-responders' pTR-Ts. **E** Distribution of pTR-T clones in post-treatment samples. Post-treatment cells with identical TCRs were mapped onto the UMAP. **F** Proportional change in phenotypic clusters in pTR-Ts after ICI treatment in responders (*n* = 4) and non-responders (*n* = 4). (two-sided paired *t* test). **G** Proportion of TRTPBL marker (CD45RA−, HLA-DR+, CD49a+, CD49b+) positive cells among total CD8+ cells in pre- and post-treatment samples (two-sided unpaired Wilcoxon rank-sum test). *RP* responder; *NRP* non-responder; *ICI* immune checkpoint inhibitors.

marker-positive cells, which are typically associated with more activated states, was significantly lower in responders than in non-responders in post-treatment samples (Fig. 4G). Collectively, our data revealed a significant phenotypic difference in pre-treatment circulating pTR-Ts between responders and non-responders. Moreover, the differentiation of circulating pTR-Ts toward a stem-like phenotype in response to ICIs is associated with a favorable prognosis.

## Validation of tumor reactivity and phenotypic changes in circulating TR-T cells following anti-PD-L1 treatment in a mouse model

To verify whether our surface markers reliably identified true TR-Ts, we used a mouse model with an artificial antigen system. In addition, we aimed to confirm whether the transition from an activated to an effector phenotype after ICI treatment specifically occurs in TR-Ts, and

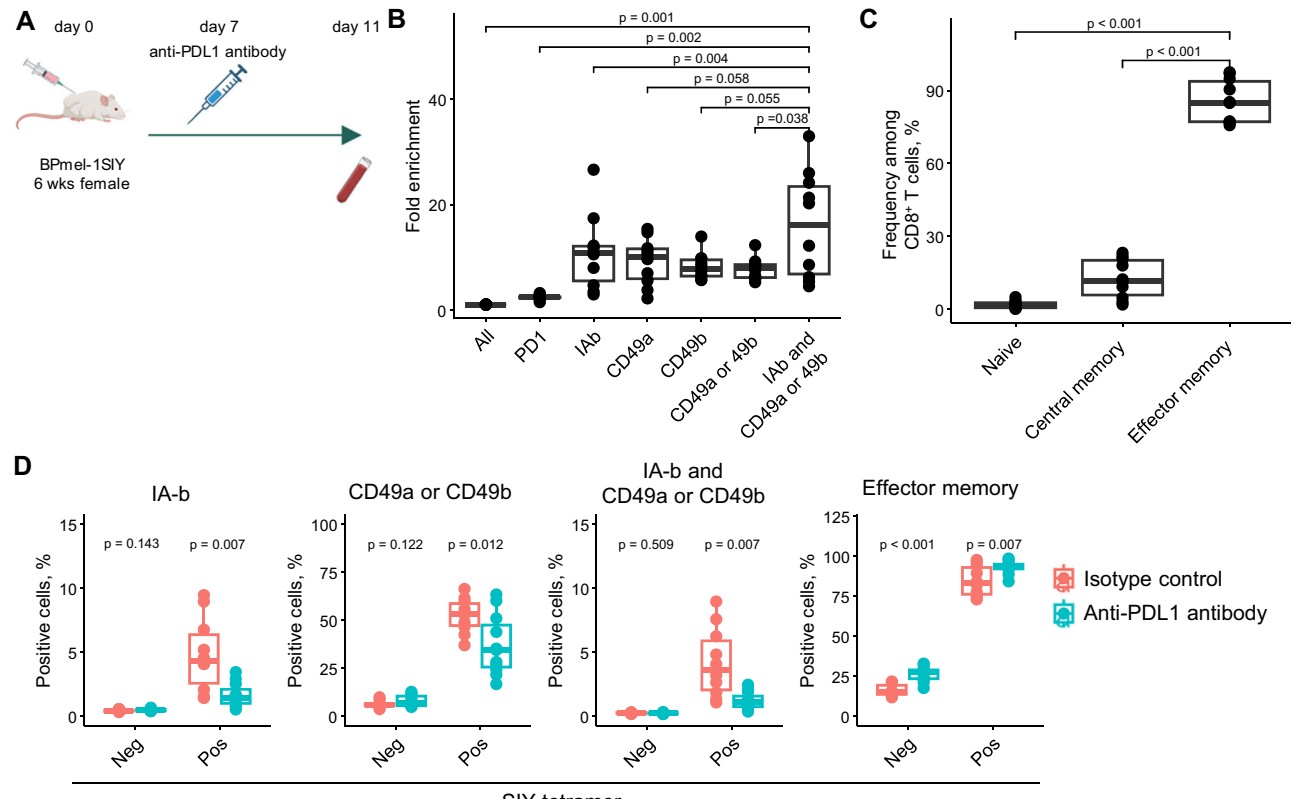

**Fig. 5 | Validation of phenotypic changes in circulating TR-T cells following anti-PD-L1 treatment in a mouse model. A** Six-week-old female mice were inoculated with a murine melanoma cell line expressing the artificial antigen SIY (BPmel-1-SIY), treated with anti-PD-L1 antibody (80 µg/mouse) on day 7, and euthanized on day 11. Created in BioRender. Ito, K. (2025) https://BioRender.com/3h5bwsu. **B** Fold enrichment of SIY tetramer-positive cells based on PD-1⁺, and circulating pTR-T marker-positive (I-Ab [MHC class II], CD49a, CD49b) CD8⁺ T cells (*n* = 10, two-sided paired *t* test). **C** Phenotype of circulating SIY tetramer-positive CD8⁺ T cells: naïve (CD62L⁺, CD44⁻), central memory (CD62L⁺, CD44⁺), and effector memory (CD62L⁻, CD44⁺) (*n* = 10, two-sided paired *t* test) **D** Therapy-induced changes in marker expression in SIY tetramer-positive and -negative cells (anti-PD-L1 antibody, *n* = 11; isotype control, *n* = 10) (two-sided unpaired *t* test). SIY: SIYR-YYGL; MHC: major histocompatibility complex; ns: not significant. In the box plots, the center line indicates the median; the box represents the interquartile range (25th–75th percentiles); whiskers extend to the most extreme values within 1.5 × the interquartile range.

whether these changes can be detected based on surface protein expression. A H-2Kᵇ-restricted antigen SIY (SIYRYYGL) was over-expressed in BPmel-1 (Bpmel-1-SIY), a murine melanoma cell line with a BRAF-mutated/PTEN-loss genotype[44,45]. The parent cell line was unresponsive to PD-1 blockade therapy but became responsive after SIY overexpression[46,47]. This suggests that SIY tetramer⁺ cells (SIY-reactive T cells) represent the predominant, if not the only, TR-T population in this model. Six-week-old female mice were inoculated with Bpmel-1-SIY (Fig. 5A). Anti-PD-L1 antibody was administered on day 7. On day 11 after tumor inoculation, circulating SIY-specific CD8⁺ T cells detected using the SIY/H2kb-tetramer assay were enriched in IAb (MHC class II)-, CD49a-, and CD49b-positive cells (Fig. 5B and Supplementary Fig. 9). The combination of IAb, CD49a, and CD49b markers allows us to concentrate the frequency of circulating SIY-specific T cells from 1.6% (0.4–5.0) to 15.3% (9.9–24.7) on average (min−max). Circulating TR-T cells predominantly exhibited an effector memory (CD44⁺ and CD62L⁻) phenotype (Fig. 5C). We next assessed the changes in these markers after treatment with the anti-PD-L1 antibody. The expression of IAb and CD49a/b in circulating SIY-tetramer⁺ T cells was significantly decreased with anti-PD-L1 antibody treatment, whereas anti-PD-L1 antibody did not affect these surface markers in tetramer-negative cells (Fig. 5D). In contrast, the proportion of effector memory T cells increased in tetramer-positive and tetramer-negative CD8⁺ T cells (Fig. 5D). Thus, our human and mouse data demonstrated that decreased TRTPBL marker expression is a unique effect of the anti-PD-L1 antibody on circulating TR-Ts, whereas effector memory differentiation occurs on a

broad range of T cells. Collectively, we validated the enrichment power of the TRTPBL marker and TR-Ts' phenotypic change following PD-1 blockade therapy in the mouse model with the artificial antigen.

## TRTPBL marker dynamics and CD38 Expression predict response to ICI

Finally, we tested the clinical relevance of our TRTPBL marker in predicting ICI response. Pre- and post-treatment blood samples were collected from 70 patients who received ICI-containing regimens for metastatic NSCLC (Fig. 6A and Supplementary Table 4). We quantified TRTPBL marker-positive cells among TCRα/β⁺CD8⁺ cells (Fig. 6B). Pre-treatment levels of marker-positive cells were not significantly different between responders and non-responders (Fig. 6C). We assessed the expression of CD38 in TRTPBL marker-positive cells, as CD38 was highly expressed in pre-treatment pTR-Ts in non-responders, as shown in Fig. 4D. The proportion of CD38-positive cells among TRTPBL⁺ CD8⁺ T cells was negatively correlated with the clinical response (Fig. 6D and Supplementary Fig. 10A). We further assessed whether changes in the proportion of TRTPBL marker-positive cells from pre-treatment to post-treatment were associated with the clinical response as shown in Fig. 4G and Fig. 5D. Patients with a decreased fraction of marker-positive cells after ICI treatment showed better response and prognosis (Fig. 6E and Supplementary Fig. 10B). Although we were unable to track the phenotype of each clonotype, we reasoned that the decrease in TRTPBL marker-positive cells was due to their differentiation from an activated state to an stem-like state (Figs. 4E–G, 5D).

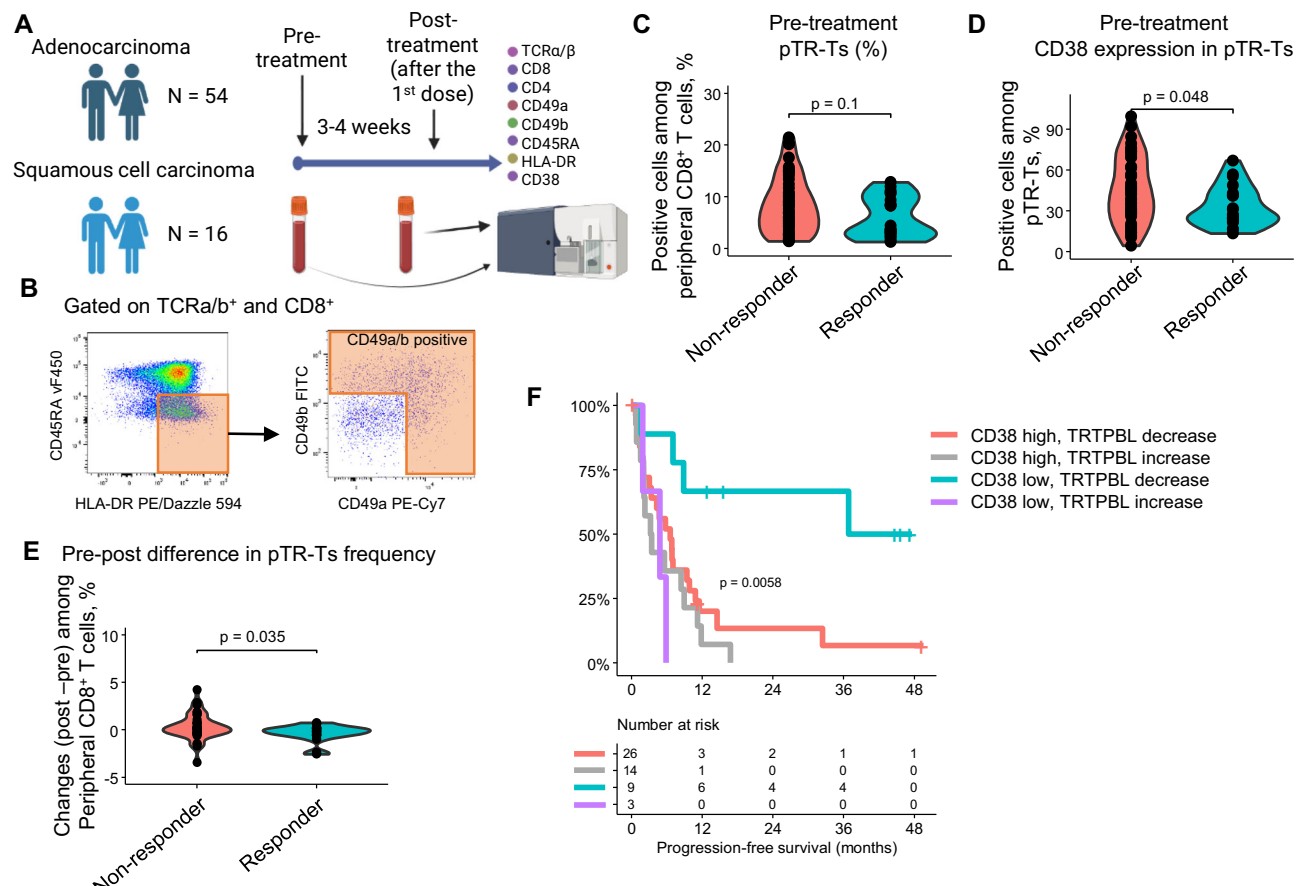

**Fig. 6 | Dynamics of circulating TRTPBL markers and CD38 expression predict ICI response. A** Seventy patients with NSCLC treated with an ICI-based regimen were enrolled. Pre- and post-treatment PBMCs were analyzed. Created in BioRender. Ito, K. (2025) https://BioRender.com/gltm5m5 **B** Gating strategy in flow cytometry to quantify TRTPBL marker-positive cells (**C**) Proportion of TRTPBL marker+ (CD45RA−, HLA-DR+, CD49a+/b+) CD8+ T cells among total CD8+ T cells was not associated with ICI response (responder, $n = 21$; non-responder, $n = 49$, two-sided unpaired Wilcoxon rank-sum test). **D** Proportion of CD38-positive cells among TRTPBL marker+ CD8+ T cells was lower in responders (two-sided unpaired Wilcoxon rank-sum test). **E** Changes (from pre-treatment to post-treatment) in TRTPBL marker+ CD8+ T cells differed between responders and non-responders. Marker+ cells significantly decreased in responders. (responder, $n = 19$; non-responder, $n = 46$, two-sided unpaired Wilcoxon rank-sum test). **F** Progression-free survival stratified by CD38 expression in pre-treatment TRTPBL marker+ cells and post-treatment changes in TRTPBL marker+ cell frequency (log-rank test). NSCLC: non-small cell lung cancer.

Finally, we combined these two markers (CD38 and TRTPBL marker+ cell reduction) to improve the predictive power. Patients with fewer CD38+ pTR-Ts and TRTPBL marker+ cells after the 1st dose of ICI showed significantly better progression-free survival (Fig. 6F). The median progression-free survival was 3.06 years in patients with CD38 low, TRTPBL marker+ decreased subgroup, compared to 0.36 years in other groups ($p = 0.0064$, log-rank test).

## Discussion

Despite the critical role of peripheral blood as a source of TR-Ts, phenotypes of circulating TR-Ts and their association with antitumor immunity have not been evaluated. Using comprehensive single-cell transcriptome, TCR, and surface protein analyses, we identified markers (CD49a, CD49b, and HLA-DR) for circulating TR-Ts in patients with NSCLC. Previous studies have focused primarily on TR-Ts in TILs, demonstrating that these cells express exhaustion and tissue-resident markers such as CD39, CD103 and CXCL13[4–7,48,49]. However, it remains unclear how these TR-Ts in TILs respond to PD-1 blockade. In this context, our study provides evidence that circulating TR-Ts in responders undergo transcriptional reprogramming following PD-1 blockade, shifting toward a previously uncharacterized population of stem-like effector memory T cells. This finding is in line with the previous report that tumor-reactive clones persist for a long time after ICIs in the peripheral blood of the responders[50]. Our study suggests that

peripheral blood contains functional, long-lived T cells that may contribute to antitumor effects during PD-1 blockade therapy.

Our comprehensive surface protein panel identified two integrins, CD49a (α1β1) and CD49b (α2β1), as markers of circulating pTR-Ts. These integrins bind to collagen types IV and I[51–53]. T cells in the tumor microenvironment do not always express these markers simultaneously[37,54]. In our non-metastatic NSCLC and melanoma dataset, expressions of either CD49a or CD49b showed higher predictive values than CD103 alone or in combination with HLA-DR (Fig. 2I). Prior studies identifying CD103+ as a marker of TR-Ts were conducted in advanced-stage melanoma, breast cancer, or colorectal cancer[16,24] which raises the possibility that tissue-residency marker expression may partly vary depending on cancer type and disease stage. This highlights the importance of combining these markers for the detection of extremely rare circulating TR-Ts.

Our trajectory analysis indicates that circulating pTR-Ts are less differentiated and less exhausted than tumor-infiltrating pTR-Ts. This finding supports the idea that peripheral organs such as draining lymph nodes serve as a reservoir of TR-Ts[55–57]. Interestingly, our data suggest that circulating pTR-Ts have already acquired a tissue-resident phenotype before infiltration. Emerging evidence suggests the presence of "circulating tissue-resident memory progenitors" that preferentially infiltrate inflamed tissues to become tissue-resident memory T cells[58]. In a mouse tumor model, once antigen-reactive

T cells infiltrate the tumor, these T cells are rarely recirculated[59]. In contrast, other researchers have shown a high proportion of TR-Ts egress from tumors via lymphatic vessels[56,60,61]. It should be noted that trajectory analysis does not provide definitive evidence of a differentiation pathway, and it remains unclear whether the phenotypic differences between circulating and intratumoral TR-Ts arise from spatiotemporal lineage relationships or from exposure to tumor antigens and the microenvironment. Future studies must elucidate how circulating TR-Ts acquire tissue-resident characteristics and contribute to tumor infiltration and retention.

Adoptive transfer therapy using autologous TILs has demonstrated clinical benefits in heavily pretreated melanoma patients and was approved by the US FDA in 2024[62]. However, the need for tumor tissues to isolate TILs limits their broader application, as tumor samples are often inaccessible for many cancer types. In this context, the peripheral blood represents a promising alternative reservoir for TR-Ts. Furthermore, TR-Ts within the tumor microenvironment are frequently functionally compromised because of chronic antigen exposure[63,64]. Our findings indicate that circulating TR-Ts are less exhausted and retain a heightened capacity to infiltrate tumors, making them a compelling source of potent and youthful TR-Ts. Several studies have introduced strategies to enrich TR-Ts from peripheral blood using autologous or artificial antigen-presenting cells[65–69]. However, a significant drawback of these traditional methods is their reliance on a priori knowledge of antigens, such as Melan-A or gp100, which restricts their utility. Given that shared tumor antigens are rare, except in cancers such as melanoma, our TRTPBL marker-based enrichment approach offers a broadly applicable alternative that does not require prior identification of tumor-specific antigens. Notably, circulating TR-Ts remain a relatively small population even after TRTPBL marker–based enrichment. Further optimization of enrichment and expansion protocols will be essential to achieve sufficient yield for clinical application. Nevertheless, our findings highlight that peripheral blood can serve as a feasible and antigen-agnostic source of TR-Ts with translational potential.

Our study highlights the potential of using circulating TR-Ts as predictive biomarkers of ICI responses. Although numerous blood-based T cell biomarkers have been reported[70], most circulating T cells are not tumor-reactive, limiting their relevance in the prediction of ICI responses. Our approach focused on circulating TR-Ts, which are more directly linked to ICI efficacy. Our data indicate that the frequency of circulating TR-Ts doesn't necessarily correlate with that of tumor-infiltrating TR-Ts or with clinical responses to ICI (Supplementary Fig. 3C and Fig. 6C). Similarly, a previous report compared the frequency of circulating tumor-reactive T cells between responders and non-responders to neoadjuvant ICI therapy in colorectal cancer; however, it was unable to leverage this frequency for response prediction[18]. Therefore, instead of focusing on cell frequency, we characterized the circulating TR-Ts based on their gene expression and surface protein profiles. We observed that a lower fraction of CD38+ pTR-Ts before treatment correlated with a better response. CD38 is a key NAD+ glycohydrolase that modulates intracellular NAD+ levels and influences T-cell activation, differentiation, and metabolic processes. Although the functional role of CD38 in response to ICI is largely unknown, the upregulation of CD38 indicates that circulating pTR-Ts in non-responders may be metabolically altered and predisposed to exhaustion[71,72]. In addition, a significant decrease in TRTPBL marker-positive cells after the first ICI dose was observed in responders, further supporting the idea that the TR-T's transition toward stem-like effector memory state is a key feature of successful treatment. A previous study has suggested phenotypic changes in circulating TR-Ts in response to ICI, although these were observed in one single clonotype from a responder patient, and that clone was not present prior to the ICI treatment[27]. In contrast to our findings, the previous report demonstrated that the TR-T became more activated from a memory

phenotype at week 4 relative to week 2. This discrepancy may reflect differential responses of pre-existing and de novo TR-Ts to ICIs, which warrants further investigation. The combination of these markers improved the predictive performance, with patients exhibiting a lower fraction of pre-treatment CD38+ pTR-Ts and a decline in TRTPBL marker-positive cells after 1st dose of therapy showing significantly prolonged progression-free survival. Given that repeated tumor biopsies are often impractical, our findings suggest that monitoring TR-T dynamics in the blood could serve as a minimally invasive and clinically relevant strategy for predicting and assessing ICI response.

This study has several limitations. First, we did not experimentally validate the tumor antigen specificity of pTR-Ts. Comprehensive functional testing of TCR antigen specificity is technically challenging and practically infeasible for large numbers of clones. Instead, we utilized a gene signature–based TR-T prediction approach, which has demonstrated excellent predictive performance in prior studies. External validation in a melanoma dataset further supports the robustness of our findings. Future studies will be needed to test the generalizability of these markers across different cancer stages and tumor types. Second, the predictive performance of variables associated with TRTPBL markers was modest and not sufficient for immediate clinical application. Given the limited number of patients, our single-cell analysis identified only a few potential biomarker candidates among ICI-treated individuals. Nonetheless, our study underscores the biological relevance of circulating TR-Ts to ICI response, suggesting that further investigations focusing on this minor PBMC compartment may help identify new predictive biomarkers.

## Methods

### Study approval
All human studies were conducted in accordance with the Declaration of Helsinki and were approved by the Institutional Review Board of Kyoto University Graduate School and Faculty of Medicine (approval number G1012). Written informed consent was obtained from all the participants.

All animal experiments were approved by the Animal Research Committee of Kyoto University and were performed in accordance with institutional guidelines.

### Patient and sample collection
We enrolled nine patients with NSCLC without EGFR mutations and one patient with melanoma who underwent surgical resection, and 70 patients with metastatic NSCLC treated with ICI therapy. Written informed consent was obtained from all the participants. All human studies included both male and female participants, unless otherwise stated. For the patient cohort with NSCLC, sex was recorded and used for sample stratification. Assigned sex determined by the national health insurance was used for the analysis.

The resected tumor specimens were examined macroscopically under the supervision of a pathologist. Tissues were mechanically dissociated into approximately 1 mm³ fragments and enzymatically digested using Tumor & Tissue Dissociation Reagent (BD Horizon) with gentleMACS Dissociator (Miltenyi Biotec) for 5 min. The cell suspension was diluted with 1% bovine serum albumin (BSA)/2 mM EDTA in phosphate-buffered saline (PBS), centrifuged at $300 \times g$ for 10 min, filtered through a 40 μm strainer, and washed twice with the same buffer. CD45 + leukocytes were enriched using CD45 MicroBeads (Miltenyi Biotec), according to the manufacturer's protocol. Isolated cells were cryopreserved in CELLBANKER (Nihon Zenn-yaku) until further use.

Peripheral blood samples (14 mL) were collected into EDTA tubes before surgery, or before and after treatment (after the first ICI dose). After centrifugation at $850 \times g$ for 10 min, the plasma was removed and stored. The remaining cells were diluted in 2% FCS/PBS and layered onto LeucoSep tubes (Greiner). PBMCs were isolated by centrifugation

at $1000 \times g$ for 10 min, washed twice, and cryopreserved in CELLBANKER.

## Single-cell library preparation

Thawed cryopreserved cells were washed using CTL Anti-Aggregate Wash (Cellular Technology Ltd.) and stained with 7-AAD (TONBO Biosciences), anti-CD45 PE (BioLegend, 2D1), and anti-CD8 APC (Bio-Legend, PRA-T8) on ice for 40 min. Cells were washed using the Curiox system (Tomy Digital Biology) at $10 \mu L/s \times 7$ cycles. CD45$^+$CD8$^+$ cells (and total live CD45$^+$ cells for TS11 and TS20 due to low yield) were sorted using FACS Melody (BD Biosciences). For the TRTPBL marker-enrichment experiment, anti-CD45RA vF450 (HI100, TONBO Biosciences), anti-CD49b FITC (P1E6-C5, BioLegend), anti-CD49a (TS2/7 BioLegend), and anti-HLA-DR PE/dazzle 594 (L243, BioLegend) were added to the panel to enrich TRTPBL marker$^+$ cells. For the melanoma dataset, MART1 tetramer staining (HLA-A*02:01 Mart-1 Tetramer-ELA-GIGILTV-PE, MBL) was used to enrich the MART1-reactive T cells.

For CITEseq, the cells were stained with TotalSeq-C human universal antibody cocktail (BioLegend) for 30 min, followed by 25 washes using the Curiox system. Single-cell suspensions were loaded into the Chromium Controller (10x Genomics) and processed using Chromium Next GEM Single Cell 5′ Reagent Kits v2. Libraries for gene expression, TCR, and surface proteins were prepared according to the manufacturer's instructions. The target read depths per cell were 20,000 for gene expression and 5000 for TCR and CITEseq.

## Single-cell RNA sequence data preparation

Reads were aligned to the GRCh38 reference genome using Cell Ranger (v7.0.1, 10x Genomics). To normalize across libraries, read lengths were trimmed to 26 bp for Read1 (barcodes) and 90 bp for read2 (alignment). Ambient RNA was removed using SoupX (v1.6.2), with a maximum contamination rate (rho) capped at 5%. Cells with poor quality metrics defined as mitochondrial gene percentage > median + 3 × mean absolute difference (MAD) or total counts and features < median − 5 × MAD were excluded. Non-T-cells lacking TCR sequences and multiplets (> 2 TCRα or β sequences) were excluded.

Data were loaded into Seurat objects (v4)[73] and normalized using log transformation. PBMC and TIL datasets were integrated separately using Harmony (v1.2.0)[74]. For TILs of TS20, CD8$^+$ clusters were manually selected using CD8A gene expression and TotalSeq-CD8 protein expression. CITE-seq data were normalized and denoised using the DSB package (v1.0.3)[75].

## Single-cell RNA sequence data analysis

Dimensionality reduction was performed using the RunUMAP and FindNeighbor functions for the harmony-integrated embeddings. Cell clustering was conducted using the FindClusters function with a resolution parameter of 1.5 to achieve fine-grained stratification. In the TIL dataset, clusters with similar expression profiles were merged manually to form interpretable cell subsets. The final cluster identities were determined based on DEGs identified using the FindAllMarkers function. The TCR genes and genes associated with polymorphisms were excluded from the analysis. DEG and DEP analysis were performed using the FindMarkers function to compare pTR-Ts with other cells. To quantify the consistency of differential expression across samples, we defined a Consistent score for each gene as follows:

$$\text{Consistent score} = \text{sign}(\text{avg\_log FC}) * \frac{1}{N} \sum_{i=1}^{N} \text{sign}(\log FC_i), \quad \text{sign}(x) \in \{-1, 1\}$$

where N denotes the number of samples, avg_logFC is the average log fold change across samples, and logFCi is the log fold change of the gene in sample. Under this definition, the score ranges from −1 to +1. A score close to +1 indicates that all samples show a direction of change that is consistent with the overall average direction. Values near 0

indicate mixed or contradictory directions among samples. Genes with a Consistency score > 0.33 (i.e., consistent direction in at least two-thirds of samples) were retained for gene enrichment analysis and pTR-T gene signature score. Single-cell datasets from melanoma patients and the ICI-treated NSCLC cohort were mapped onto the reference UMAP generated from the NSCLC surgical cohort using the FindTransferAnchors and MapQuery functions in the Seurat package (v4)[76].

T cell clonotypes were defined based on the complete nucleotide sequences of the CDR3 of both the α and β TCR chains, including the corresponding V, (D), and J gene segments. Cells were considered clonally related only if they shared identical sequences and gene usage in both chains. Cells with only one productive chain (α or β) were treated as independent clonotypes and were not merged. VR-Ts were identified by matching TCR sequences to entries in the VDJdb database[32]. CDR3 sequences against MHC class I-restricted epitopes were selected. A match in either the α or β chain to known viral CDR3 sequences (i.e., CMV, EBV, or influenza) was considered sufficient for annotation as VR-Ts, as both chains were not always available in the database.

To identify TR-Ts, we used the MANAscore described by Zeng et al.[31]. For each clonotype, we calculated the MANAscore using the UCell algorithm (v2.0), which is based on the Mann–Whitney U statistic and is robust to sequencing depth and dropout[77]. Although the original study used 5 cells as a cut-off to define TR-Ts, we set the cut-off value at ≥ 5% of positive cells within a CD8$^+$ T cell clone to address the cell number unbalance. Clonotypes which exceeded this threshold and accounted for more than 1% of all CD8+ TILs were designated as pTR-Ts.

To create the pTR-T signature score, we extract genes with positive log2 fold changes and the consistency score > 0.33 from the DEGs between circulating pTR-Ts vs. other cells. The pTR-T signature score and other scores were calculated using the UCell algorithm.

Trajectory analysis was performed using the Monocle3 package (v1.3.1)[78]. Cells belonging to the pTR-T clonotype were extracted from the PBMC and TIL datasets. After batch correction for patient identity and sample type (PBMC vs. TIL) using Harmony, dimensionality reduction via UMAP and unsupervised clustering was performed. Pseudotime trajectories were inferred to determine differentiation states and lineage relationships between circulating and tumor-resident pTR-Ts.

## In vitro expansion of human melanoma TILs

TILs were expanded as described previously[63] from a primary lesion of a patient with HLA-A*0201 positive melanoma, MELC6. In brief, melanoma tumor digests were initiated in 2-mL wells containing RPMI 1640 supplemented with 10% human serum, antibiotics, and recombinant human IL-2 (6000 IU/ml; PeproTech Inc.) in a humidified 37 °C incubator with 5% CO2. After 5 days, half of the medium was aspirated from the wells and replaced with fresh complete medium and IL-2; this was repeated every 2–3 days thereafter, as needed.

## Enrichment and cloning of MART1-specific CTLs and identification of TCR clonotypes

The expanded TILs derived from MELC6 ($1 \times 10^6$ cells per well) were stimulated with 100-Gy irradiated HLA-A*0201 positive thymoma cell line T2 cells ($1 \times 10^6$ cells per well) under TIL culture conditions. Irradiated T2 cells were pre-pulsed with an HLA-A*0201-restricted MART1 peptide (EAAGIGILTV) at $1 \mu M$ for 1 h before the stimulation. The stimulation was repeated 3 times every other week. Enrichment of MART1-reactive CTLs was confirmed using the MART1 tetramer according to the manufacturer's instructions. MART1-specific CTL clones were established from stimulated TIL, as described previously[38]. Clonality of expanded T cells was verified using TCRβ-chain V region analysis based on the 5′ RACE method as described previously[38].

### CD8[+] T cell infiltration quantification

To assess CD8[+] T cell infiltration in tumor tissues, FFPE sections from each patient were stained with an anti-CD8 antibody (VENTANA, clone SP57). Two tissue sections were analyzed for each case. Digital image analysis was performed using QuPath (version 0.2.3), open-source software for quantitative pathology[79]. Within each slide, three representative tumor areas were manually selected by a trained operator. CD8[+] T cell densities were quantified in each selected region, and the average values across the three regions were used for subsequent analyses.

### Mouse experiment

Mice were maintained in a specific pathogen-free (SPF) animal facility under a 12 h light/dark cycle with ad libitum access to food and water. Experimental and control mice were co-housed whenever possible. Mice were monitored daily, and humane endpoints were applied when animals exhibited > 20% body weight loss, impaired mobility, ulceration, or when tumor volume exceeded 1000 mm$^3$, at which point mice were euthanized.

Fourteen six-week-old female C57BL/6 N mice (Clea Japan) were randomly assigned to control or treatment groups ($n = 10$ per group). Mice were inoculated intradermally on the right flank with $1 \times 10^6$ BPmel-1-SIY cells, a murine melanoma cell line engineered to express the SIY (SIYRYYGL) epitope via lentiviral transduction of three tandem SIY minigenes under a puromycin-selectable cassette, as previously described[45].

On day 7 post-inoculation, mice received a single intraperitoneal injection of either anti−PD-L1 monoclonal antibody (clone 1-111 A.4, 80 μg/mouse)[80] or isotype control (rat IgG2a, κ; Bio X Cell). Peripheral blood was collected on day 11 via facial vein puncture (live sampling) or from the vena cava after euthanasia in a CO2 chamber.

### Flow cytometry

**Mouse samples.** Peripheral blood (100 μL) was collected from the mice and lysed using 1 mL of VersaLyse buffer (Beckman Colter) for 10 min at room temperature. Cells were washed twice with PBS containing 2% fetal calf serum (FCS) and blocked with anti-CD16/32 Fc blocker (93, BioLegend, 1%). SIY-specific CD8[+] T cells were stained with SIY tetramer (T-Select H-2Kb Negative (SIY) Tetramer, MBL, 2.5% v/v) and incubated for 20 min at room temperature. After washing and diluting in Brilliant Stain Buffer (BD Biosciences), the cells were stained with the following fluorochrome-conjugated antibodies for 20 min at room temperature:

- CD45 BUV395 (30-F11, BD Horizon), 0.2%
- TCRβ BUV737 (H57-597, BD Horizon), 1%
- CD8 AF647 (KT15, MBL), 0.625%
- CD4 AF700 (RM4-5, BioLegend), 0.5%
- CD49a PE-Cy7 (HMa1, BioLegend), 1%
- CD49b PerCP/Cy5.5 (HMa1, BioLegend), 1%
- PD-1 BV605 (29 F.1A12, BioLegend), 1%
- MHC-II (I-A/I-E) FITC (M5/114.15.2; BioLegend), 1%
- CD62L BV711 (MEL-14, BioLegend), 0.5%
- CD44 BV785 (IM7, BioLegend), 0.5%
- 7-AAD (TOMBO Biosciences), 0.2%

Stained samples were washed twice with 2% FCS/PBS and analyzed using an ID7000 spectral flow cytometer (Sony Biotechnology).

**Human samples.** Cryopreserved PBMCs were thawed in a 37 °C water bath and slowly diluted in warm, complete RPMI medium containing 30 U/mL DNase I (Sigma, D5025). After centrifugation ($250 \times g$ for 7 min), the cells were washed and counted. For staining, $1 \times 10^5$ PBMCs were incubated with human IgG (FUJIFILM) as an Fc blocker (1% v/v) for 30 min at 4 °C, followed by adding the antibody cocktail without washing.

The following fluorochrome-conjugated antibodies were used:
- TCRα/β BUV737 (IP26, BD OptiBuild), 1%
- CD8 BUV395 (RPA-T8, BD Horizon), 0.125%
- CD4 APC-Cy7 (RPA-T4, TONBO Biosciences), 0.5%
- CD49a PE-Cy7 (TS2/7, BioLegend), 2%
- CD49b FITC (P1E6-C5, BioLegend), 2%
- CD45RA vF450 (HI100, TONBO Biosciences), 0.5%
- HLA-DR PE/Dazzle 594 (L243, BioLegend), 2%
- CD38 BV711 (HIT2, BioLegend), 2%
- 7-AAD (TONBO Biosciences), 1%

Stained samples were washed twice with 2% FCS/PBS and analyzed using an ID7000 spectral flow cytometer (Sony Biotechnology).

### Statistical analysis

All statistical analyses were performed using the R software (version 4.2.2). Pearson's correlation coefficient was calculated to assess the linear relationship between variables. For comparisons between the two groups, Student's *t* test and Mann−Whitney U-test were used for variables with or without normal distribution, respectively. Statistical significance was set at a two-tailed *p*-value < 0.05. Details of the exact *p*-values, sample sizes, and test types are provided in the corresponding figure legends.

ROC curve analysis and AUC calculations were performed using the pROC package version 1.18.5. The AUC values were calculated without cross-validation, and the cut-off thresholds for classification were determined using the Youden index to maximize sensitivity and specificity.

Progression-free survival was analyzed using the Kaplan–Meier method. Differences between the survival curves were assessed using log-rank tests. Cut-off points for stratifying patients based on biomarker expression (e.g., CD38 positivity and TRTPBL dynamics) were determined using time-dependent ROC analysis, with the concordance index (C-index) used to identify the optimal threshold at 1 year.

All plots were generated using the ggplot2, survival, and survminer packages in R. No data were excluded from the analysis, unless explicitly stated.

### Reporting summary

Further information on research design is available in the Nature Portfolio Reporting Summary linked to this article.

## Data availability

The single-cell RNA sequencing data generated in this study have been deposited in the NCBI Sequence Read Archive (SRA) under the BioProject accession number PRJNA1415239. The single-cell data generated in this study and all source data supporting the findings of this study have been deposited in the Zenodo database (https://doi.org/10.5281/zenodo.17342100)[81]. The raw sequence data are protected and are available from the corresponding author upon reasonable request and ethical approval due to consent from patients. Source data are provided in this paper.

## Code availability

Scripts supporting the data were based on existing computational tools. The code used for the analysis are available in https://doi.org/10.5281/zenodo.17342100 or https://github.com/itokatsu0320/pTR-T.

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

## Acknowledgements

We are grateful to Prof. T. Honjo for founding the Center for Cancer Immunotherapy and Immunobiology, where this research was conducted, and for his valuable comments. We thank the Center for Anatomical, Pathological and Forensic Medical Research in the Graduate School of Medicine, Kyoto University, for supporting histological analyses, and the staff of the Clinical Bio-Resource Center in Kyoto University Hospital for harvesting human samples. This work was supported by JSPS KAKENHI No. JP24K02325 (T.Y.), JP21H03087 (K.C.), JP24K22068 (T.Y.); AMED under grant No. JP23zf0127004 (K.C.) and JP24ama221330 (K.C.); Menarini Biomarkers Singapore Pte Ltd (Tasuku Honjo), Yanai Fund (Tasuku Honjo), Meiji Holdings Co., Ltd. (Tasuku Honjo, K.C.), Meiji Seika Pharma Co., Ltd. (Tasuku Honjo, K.C.). We sincerely thank H. Nakajima, M. Kiyono, M. Esaki, Y. Kitawaki, Y. Haku, K. Kitaoka, W. Jun, M. Shimazaki and the Lab members for assistance with sample preparation.

## Author contributions

Katsuhiro.I., T.Y., and K.C. designed the research; Katsuhiro.I., K.M., Tomoko.H., M.ML.L., S.K., K.A., and Y.S. conducted experiments; Kei.I., T.K., T.M., H.D., H.O., H.Y., Toyohiro.H., S.K., K.A., Y.S., and T.I. acquired data; Katsuhiro.I. and T.I. analyzed data; and Katsuhiro.I., T.Y., Tomoko.H., and K.C. wrote the manuscript; T.Y. and K.C. supervise the entire research.

## Funding

Chugai Foundation for Innovative Drug Discovery Science: C-FINDs (Merrin M.L. Leong).

## Competing interests

The authors declare no competing interests.
