## [Transparent Peer Review file · Nature Communications]

Phenotype of Circulating Tumor-Reactive T Cells Predicts Immune Checkpoint Inhibitor Response in Non-Small Cell Lung Cancer

Corresponding Author: Dr Tomonori Yaguchi

Version 0:

Reviewer comments:

Reviewer #1

(Remarks to the Author)

The goal of the present manuscript is to understand how tumor-reactive T cell programming changes as cells traffic from the periphery into tumors. They use a variety of -omics technologies to study gene and protein expression of anti-tumor T cells across several different lung cancer cohorts and validate a panel of tumor-reactive T cell peripheral blood markers. They show that shifts in marker positivity and specifically expression of CD38 can predict ICI response. There are significant concerns about the rigor in computational analyses, justification of methods, and contextualization with existing literature, thus leading to questions about the interpretation of results. Major and minor points are below.

Major

1. There is no justification provided for the inclusion of lung cancers with EGFR mutations, as these are histologically and immunologically distinct
2. It is unclear why the authors chose to use NeoTCR8 over other, more parsimonious, scores. CXCL13 has been proposed as a singular marker of tumor-reactive TIL (Liu, Nat Cancer, 2022), and the slightly better MANAScore (Zeng et al, Nat Comm, 2025) contains only 3 genes.
3. It is unclear why the 'Exhausted' cluster is not also labeled as tissue resident, considering the high expression of ITGAE. To remain consistent with reported literature, I would refer the authors to the first two studies to profile tumor-reactive TIL at single cell resolution, Oliveira et al, Nature, 2021 and Caushi et al, Nature, 2021, both of which are not cited, discussed, nor contextualized in the present study.
4. pTR-T need to be shown for each patient individually
5. One of the more salient findings is that peripheral TR-T are in a GZMKhi subset. More attention should be paid to this given that GZMK is emerging as one of the key cytokines of interest related to tumor-reactivity and ICI response.
6. The authors need to confirm that the DEGs are consistent at the patient level. Cell level DGE analyses are extremely susceptible to patient-specific effects.
7. It's unclear why the authors are developing their own pTR-T gene signature score and how it compares with other published scores.
8. It is stated that 12,876 peripheral CD8+ T cells underwent analysis in Figure S4E, but there is no indication of how many of those were multimer positive. From the figure, it looks as if only 3 cells matching the multimer+ clone were identified, which is wildly inconsistent with the expected 1.0%
9. The Results for Figure 3 claim that the Exhaustion signature is higher in TIL and the progenitor signature is higher in PBMC, however the data shown in Figure 3C show the complete opposite. The data do not support this statement.
10. The statement that "Expression of TRTPBL markers were relatively stable across clones" is not actually supported by the data. Yes, in aggregate, there is no difference in expression, however this is because most patients have either a very prominent increase or a very prominent decrease in expression of these genes, thus cancelling out any differences at the cohort level. I would encourage the authors to explore these disparate Results according to patients.
11. The authors need to provide more information regarding the metastatic cohort related to number of cells that were sequenced from each patient and the UMPA distribution of clusters for each patient.
12. The Results for Figure 4 claim to show genes that are significantly different between R and NR TR-T, however only 2 genes actually have an adjusted p-value <0.05. If the authors insist, these genes should also be confirmed at the patient level to ensure the differences in gene expression are not driven by patient-specific effects
13. The authors need to validate that the Results shown in Fig. 4G are consistent across patients

Minor

1. There is no table provided showing the number of cells that were sequenced from each individual sample, making it difficult to assess the generalizability of the results
2. Most of the CMV-specific clones in the public databases are derived from CD4+ T cells. There can be quite a lot of overlap in Vbeta sequences between CD4 and CD8 T cells despite recognizing distinct antigens. Although the Methods indicate that "clones" were identified based on alpha and beta sequences, most public databases only contain beta sequences. The authors need to clarify this point.
3. This study seems to contradict prior findings that flu-specific T cells in lung cancers are of a tissue-resident memory phenotype. This point is not addressed.
4. In Figure 2I, "Rami et al" should be "Yossef et al"
5. Lines 342-351 seemingly come out of nowhere. It is unclear how this relates to the data that were discussed in the prior sentences.
6. In general, this study fails to report the findings within the context of prior work on tumor-specific TIL in lung cancer.

Reviewer #2

(Remarks to the Author)

The majority of T cells in patients with cancer are not specific to tumor antigens, so unravelling the biology of tumor specific T cell responses, as well as developing predictive tests and identifying sources of T cells for therapy, rely on the identification of tumor specific T cell subsets. In recent years, transcriptional signatures have been identified that differentiate tumor specific T cells in tumors from tumor non-specific bystander T cells, but less work has been done on the defining characteristics of these cells in the peripheral blood, where they are more accessible to analysis, possibly less functionally impaired, but significantly rare. This work uses single cell RNA sequencing to identify intratumoral CD8+ T cells with a transcriptional signature of tumor antigen reactivity in a cohort of smoking patients with non-small cell lung cancer, and then utilize the unique TCRVb gene rearrangements that mark specific T cell clones to interrogate the phenotype of these cells in the peripheral blood. They identify transcriptional and surface marker signatures enriched for these cells in the blood, find that these cells expand in the blood and change in phenotype on successful treatment with PD-1 inhibitors, and validate this subset contain cells specific to a known tumor antigen in a patient with melanoma. These include a phenotype of CD49a/b and HLA-DR that is novel to this study and may lend insight into the potential trafficking of these cells. They find that tumor specific CD8 T cells of known specificity to a model antigen in a mouse model of PD-1 inhibitor sensitive melanoma have similar characteristics. They identify pre-treatment lack of CD38 expression in this blood subset and expansion of clones in this subset as modest biomarkers of treatment response. While this is not the first description of similar cell populations, and the direct clinical consequences of using this cell population as a biomarker or source of therapeutic T cells is likely to be modest, this work is well done, a valuable contribution to the field, and the weaknesses identified are relatively minor and can be addressed by changes in emphasis in the results and discussion.

Minor points:

1-pseudotime analysis does not demonstrate a differentiation pathway (in fact it assumes one), and the authors should be cautious about over-interpreting their findings in figure 3. It is possible, maybe even likely that most of the phenotypic difference between intratumoral tumor specific cells and their counterparts in the blood is exposure to antigen in the tumor, and it is not at all clear that peripheral T cells are a source of intratumoral T cells during treatment.

2-From their data, it appears the clinical utility of analysis of these cell populations for pre-treatment or early treatment prognostication will be limited, but that is unfortunately the case with most such biomarkers, and does not detract from what these correlations suggest about the biology.

3-While it is tantalizing that peripheral blood T cell populations could be used as a source of T cells for therapeutic products, it is not clear whether these cells will be sufficiently enriched in the relevant subsets or sufficiently common to make this happen.

4-The relative lack of validation of tumor antigen specificity of the identified putative tumor specific clonotypes is a weakness, but as the authors point out, such studies are challenging and essentially impossible to do comprehensively, and the work they present in melanoma as validation is adequate for the purposes of this paper.

Version 1:

Reviewer comments:

Reviewer #1

(Remarks to the Author)

The authors have done a great job addressing the comments. Nothing further to add.

Reviewer #2

(Remarks to the Author)

The changes are adequate for publication from my standpoint.

For all Reviewers

We sincerely appreciate the reviewers' valuable comments and suggestions, which improve this manuscript. We have carefully addressed your questions point by point below. Text revised in response to Reviewer 1 is highlighted in red, revisions for Reviewer 2 in blue, and additional clarifying text in green. No other parts of the original manuscript were modified.

Reviewer #1 (Remarks to the Author):

The goal of the present manuscript is to understand how tumor-reactive T cell programming changes as cells traffic from the periphery into tumors. They use a variety of -omics technologies to study gene and protein expression of anti-tumor T cells across several different lung cancer cohorts and validate a panel of tumor-reactive T cell peripheral blood markers. They show that shifts in marker positivity and specifically expression of CD38 can predict ICI response. There are significant concerns about the rigor in computational analyses, justification of methods, and contextualization with existing literature, thus leading to questions about the interpretation of results. Major and minor points are below.

We sincerely appreciate your critical and constructive comments on our manuscript. We have carefully considered all the points raised and made substantial revisions accordingly. We believe that these thoughtful suggestions have significantly improved the rigor, clarity, and overall quality of the study. Below, we provide detailed, point-by-point responses to each of the reviewer's comments and indicate how we have addressed them in the revised manuscript.

Major

1. There is no justification provided for the inclusion of lung cancers with EGFR mutations, as these are histologically and immunologically distinct

Thank you for your comment on this point. As you correctly pointed out, tumors harboring EGFR mutations are known to exhibit distinct tumor immune microenvironments. After careful consideration, we excluded patients with EGFR mutations (n=3) from the analysis and re-performed all bioinformatic and statistical

analyses using only EGFR-wild-type NSCLC cases (n=9). Importantly, this exclusion did not lead to any significant changes in our main findings. All related figures and corresponding text in the manuscript have been revised accordingly.

(Page 10, Line 150-151 in the *Result* section of the revised manuscript)

We generated a single-cell RNA-seq/TCR-seq/CITE-seq dataset of CD8⁺ T cells isolated from resected tumor tissues and peripheral blood of nine patients with NSCLC without EGFR mutations (Figure 1A, Supplementary Table 1).

(Page 34, Line 563 in the *Method* section of the revised manuscript)

We enrolled nine patients with NSCLC without EGFR mutations

2. *It is unclear why the authors chose to use NeoTCR8 over other, more parsimonious, scores. CXCL13 has been proposed as a singular marker of tumor-reactive TIL (Liu, Nat Cancer, 2022), and the slightly better MANAscore (Zeng et al, Nat Comm, 2025) contains only 3 genes.*

Thank you for this insightful comment. We agree with your suggestion and have replaced the NeoTCR8 score with the MANAscore, as it demonstrated superior predictive performance, consistent with the report by Zeng et al. (Nat Commun, 2025). The majority of tumor-reactive T cells overlapped between the two scoring methods, indicating consistency across approaches. Notably, adopting the MANAscore did not affect any of our main findings or conclusions. All related figures and the corresponding text in the manuscript have been revised accordingly.

(Page 11, Line 166-169 in the *Result* section of the revised manuscript)

We applied the MANAscore reported by Zeng et al. (31). This score consists of two positive genes (CXCL13 and ENTPD1) and one negative gene (IL7R) and could detect TR-Ts with higher sensitivity than other RNAseq-based gene signatures.

(Revised Figure 1D, Neoantigen-reactive score (MANAScore) calculated and shown in UMAP)

3. It is unclear why the 'Exhausted' cluster is not also labeled as tissue resident, considering the high expression of *ITGAE*. To remain consistent with reported literature, I would refer the authors to the first two studies to profile tumor-reactive TIL at single cell resolution, Oliveira *et al*, *Nature*, 2021 and Caushi *et al*, *Nature*, 2021, both of which are not cited, discussed, nor contextualized in the present study.

Thank you for this important comment. You are absolutely correct that the “Exhausted” cluster exhibits high expression of *ITGAE*, a key molecule associated with tissue-resident memory (TRM) T cells. To maintain consistency with previously reported literature, we have revised the cluster nomenclature accordingly, renaming the “Exhausted” cluster as the “TRM(2)/Exhausted” cluster. We have also cited and discussed the key studies by Oliveira *et al*. (*Nature*, 2021) (Ref #28 in the manuscript) and Caushi *et al*. (*Nature*, 2021) ((Ref #27 in the manuscript)), which characterized the phenotype of tumor-reactive T cells at single-cell resolution, and we highlight the similarities between their tumor-reactive T cells and our predicted TR-Ts. These revisions have been incorporated into the *Results* section as described below.

(Page 10, line 157-158 in the *Result* section of the revised manuscript)

We labeled CD8+ TILs based on the previous studies to profile TR-Ts in NSCLC and other cancers (27, 28),

(Revised Figure 1B, High-resolution single-cell CD8⁺ TILs mapped to 8 clusters)

(Page 10-11, line 163-171 in the *Result* section of the revised manuscript)

Early studies have profiled the phenotype of TR-Ts in TILs based on experimentally confirmed antigen specificity (27, 28). In the present study, rather than focusing on validating antigen specificity, we aimed to define TR-T in a more comprehensive manner by applying a gene signature-based approach. We applied the MANAscore reported by Zeng et al. (31). This score consists of two positive genes (CXCL13 and ENTPD1) and one negative gene (IL7R) and can detect TR-Ts with higher sensitivity than other RNAseq-based gene signatures. The MANAscore was enriched in TRM(2)/Exhausted clusters of CD8⁺ TILs (Figure 1D), which is consistent with earlier findings showing that TR-Ts exhibit a tissue-resident and exhausted phenotype (27, 28).

4. *pTR-T need to be shown for each patient individually*

We agree with this comment. The distributions of pTR-Ts in TILs and PBMCs for each patient are shown individually in Supplementary Figure 1D and 3B. We also revised Figure 2C and added Supplementary Figure 3E to show cluster distribution across individual patients. pTR-Ts were detected in 8 of 9 patients in TILs and 6 patients in PBMCs with varied frequency. We observed that the distribution patterns of pTR-Ts in TILs differed among patients. To clarify these findings, we have added the following sentences to the *Results* section:

(Page 12, Line 186-187 in the *Result* section of the revised manuscript)

The distribution pattern of pTR-Ts varied among patients (Supplementary Figure 1D).

(Page 13, Line 209-211 in the *Result* section of the revised manuscript)

Circulating pTR-Ts were detected in only six patients (TS04, 13, 16, 19, 20, and 25) with varied frequency (0.2–1.1%) (Supplementary Figure 3B).

(Page 14, line 218-221 in the *Result* section of the revised manuscript)

We evaluated the cluster in which the circulating pTR-Ts were preferentially detected and found that they were enriched in the effector memory (2, 3, 8 11) and proliferating cluster (14), particularly in cluster 11 (Figure 2B, C, Supplementary Figure 3E)

(New Supplementary Figure 1D, Distribution of pTR-Ts in TILs stratified by patients)

(New Supplementary Figure 3B, Distribution of pTR-Ts in PBMCs stratified by patients)

(Previous Figure 2C)

(Revised Figure 2C, Proportion of pTR-T in cluster 11 and all CD8⁺ T cells stratified by patients.)

(New Supplementary Figure 3E, For peripheral CD8⁺ T cell analysis, the proportions of pTR-Ts in each cluster were plotted, stratified by patient)

5. One of the more salient findings is that peripheral TR-T are in a GZMK^{hi} subset. More attention should be paid to this given that GZMK is emerging as one of the key cytokines of interest related to tumor-reactivity and ICI response.

Thank you for your valuable comment. We agree that pTR-Ts are skewed toward the GZMK^{hi} cluster. As the reviewer pointed out, recent studies have shown that GZMK is a key molecule related to tumor reactivity and ICI response. To emphasize this finding, we have cited these studies and added the following sentences to the *Results* section:

(Page 14, line 225-229 in the *Result* section of the revised manuscript)

This cluster also expresses GZMK, which is emerging as a key molecule related to tumor reactivity and ICI response(Supplementary Figure 3A) (11, 33, 34). The enrichment of pTR-Ts in the GZMK⁺ cluster suggests that these cells may contribute to antitumor immunity in the peripheral blood.

6. The authors need to confirm that the DEGs are consistent at the patient level. Cell level DGE analyses are extremely susceptible to patient-specific effects.

We completely agree with this comment. As noted, the number of TR-Ts varies among patients and could influence cell-level analyses. To address this, we performed DEG analysis at the individual patient level in addition to cell-level analysis. We then created a consistency score, and DEGs and DEPs that were consistently observed in at least two-thirds of patients (highlighted in red in the revised Figure 2D and 2E, and orange in the new Supplementary Figures 4A and 4B) were selected for gene enrichment analysis, pTR-T gene signature score analysis, and TRTPBL marker detection. We believe this step significantly improves the robustness of the signature. We have added these two supplementary figures and the following sentences to the *Results* and *Methods* sections:

(Page 15, line 242-246 in the *Result* section of the revised manuscript)

Following the cell-level DEG and DEP analyses, we performed DEG and DEP analyses across individual patients to minimize patient-specific effects and used DEGs that were consistently observed in multiple patients for subsequent analyses (Figure 2D and 2E, Supplementary Figure 4A and 4B, Supplementary Table 6 and 7).

(Page 37, line 620-631 in the *Method* section of the revised manuscript)

DEG and DEP analysis were performed using the FindMarkers function to compare pTR-Ts with other cells. To quantify the consistency of differential expression across samples, we defined a *Consistent score* for each gene as follows:

Consistent score

$$= \text{sign}(\text{avg_logFC}) * \frac{1}{N} \sum_{i=1}^N \text{sign}(\text{logFC}_i), \quad \text{sign}(x) \in \{-1, 1\}$$

where N denotes the number of samples, avg_logFC is the average log fold change across samples, and logFC_i is the log fold change of the gene in sample. Under this definition, the score ranges from -1 to +1. A score close to +1 indicates that all samples show a direction of change that is consistent with the overall average direction. Values near 0 indicate mixed or contradictory directions among samples. Genes with a Consistency score > 0.33 (i.e., consistent direction in at least two-thirds of samples) were retained for gene enrichment analysis and pTR-T gene signature score.

(Revised Figure 2 (D) DEGs between circulating pTR-Ts and other CD8⁺ T cells. Genes that were consistently up- or downregulated across patients (consistency score > 0.33, as shown in Supplementary Figure 4A) are highlighted in red. (E) DEPs between circulating pTR-Ts and other CD8⁺ T cells. Proteins that were consistently up- or downregulated across patients (consistency score > 0.33, as shown in Supplementary Figure 4B) are highlighted in red.)

(New Supplementary Figure 4, (A) Differentially expressed genes between pTR-Ts and other CD8+ T cells across patients. “All” represents results aggregated from all patient samples, which correspond to the cell-level analysis shown in Figure 4D. Consistency was evaluated based on the direction of log₂ fold change, where a value of 1 indicates that the gene was regulated in the same direction (either up or down) across all patients (see Methods). (B) Differentially expressed proteins between pTR-Ts and other CD8+ T cells across patients. “All” represents results aggregated from all patient samples, which correspond to the cell-level analysis shown in Figure 4E. Consistency was calculated using the same method described in Supplementary Figure 4A.)

7. It's unclear why the authors are developing their own pTR-T gene signature score and how it compares with other published scores.

Thank you for the comment. We acknowledge that circulating pTR-T gene signatures have been published. However, the previously reported signature (NeoTCRPBL) was derived from colorectal and breast cancers, and *circulating* pTR-T gene signatures derived from NSCLC have not been reported. This motivated us to develop our own circulating pTR-T signature and compared with NeoTCRPBL. We found that only ~30% of genes overlapped between NeoTCRPBL and our signature. Our signature showed

higher discriminative value (AUC 0.87 vs. 0.75), which suggests TR-Ts may have tumor-type-specific transcriptional programs. We added the following sentences to the *Results* section:

(Page 16, line 254-259 in the *Result* section of the revised manuscript)

Although a circulating TR-T gene signature (NeoTCRPBL) has been reported (16), it was derived from melanoma, colorectal, and breast cancers, and no circulating TR-T signature has been established for lung cancer. Therefore, we developed a lung cancer-derived circulating pTR-T signature to evaluate whether the transcriptional features of circulating TR-T cells in NSCLC are comparable to those reported in other tumor types.

(Page 16, line 262-268 in the *Result* section of the revised manuscript)

We compared our circulating pTR-T signature with the NeoTCRPBL signature and found that only ~30% of genes overlapped (Supplementary Figure 4C). Our signature showed a higher area under the curve (AUC = 0.87) for identifying pTR-Ts in receiver operating characteristic (ROC) analysis compared with NeoTCRPBL (AUC = 0.75) (Figure 2G), indicating improved performance in predicting peripheral TR-Ts. These results suggest that the characteristics of pTR-Ts may be affected by tumor type.

(New Supplementary Figure 4C, Shared and distinct genes between the pTR-T gene signature and the NeoTCRPBL signature)

(New Figure 2G, ROC analysis for detecting circulating pTR-Ts based on gene signature)

score derived from the pTR-T gene signature (Supplementary Figure 4A, Supplementary Table 2) and NeoTCRPBL signature reported by Yossef et al.)

8. It is stated that 12,876 peripheral CD8⁺ T cells underwent analysis in Figure S4E, but there is no indication of how many of those were multimer positive. From the figure, it looks as if only 3 cells matching the multimer⁺ clone were identified, which is wildly inconsistent with the expected 1.0%

Thank you for your question. As you correctly pointed out, three cells in PBMCs matched the multimer⁺ clone (clone 1). While the frequency of multimer⁺ clones was approximately 1.0% among CD8⁺ TILs, their frequency in CD8⁺ PBMCs was less than 0.1%. To clarify this point, we have revised the manuscript accordingly. We also revised the labels in Supplementary Figure 5B and 5E to indicate the origin of samples.

(Page 19, line 310-311 in the *Result* section of the revised manuscript)

We detected three MART1-specific T cells (clone 1).

(Revised Supplementary Figure 5B, Frequency of MART1 tetramer-positive cells in the fresh tumor digest)

(Revised Supplementary Figure 5E, Clone 1 was projected onto the UMAP of CD8⁺ PBMCs. Clone 1 was located within the cluster 3 and 14.)

9. The Results for Figure 3 claim that the Exhaustion signature is higher in TIL and the progenitor signature is higher in PBMC, however the data shown in Figure 3C show the complete opposite. The data do not support this statement.

We sincerely apologize that the label was flipped. We corrected the label so that the Figure3C now support the claim. Thank you for pointing this out. The revised Figure 3C is shown below.

(Revised Figure 3C, Averaged clone-level exhaustion and progenitor signature scores between pTR-Ts in PBMCs and those in TILs. Each dot and line represents a clonotype.

(paired *t*-test))

10. The statement that “Expression of TRTPBL markers were relatively stable across clones” is not actually supported by the data. Yes, in aggregate, there is no difference in expression, however this is because most patients have either a very prominent increase or a very prominent decrease in expression of these genes, thus cancelling out any differences at the cohort level. I would encourage the authors to explore these disparate Results according to patients.

We appreciate the reviewer’s insightful comment. When we plotted the expression of TRTPBL markers stratified by patient, we found that changes in marker expression exhibited different patterns depending on the clone, even within the same patient. Notably, despite the different changing patterns, TRTPBL marker expression was consistently positive in PBMCs and TILs. To clarify this point, we have added Supplementary Figure 6 and revised the manuscript accordingly.

(Page 14–15, line 337–345 in the *Result* section of the revised manuscript)

In contrast, the expression of TRTPBL markers (CD49a, CD49b, and HLA-DR) was relatively inconsistent across clones (Figure 3D). Changes in marker expression exhibited distinct patterns depending on the clone, even within the same patient (Supplementary Figure 6). This heterogeneity may reflect differences in TCR affinity or in the stage of T-cell differentiation between peripheral blood and tumor-infiltrating populations. Notably, despite these variable changes in TRTPBL markers, TRM-associated and activation markers were consistently expressed at relatively high levels in both peripheral blood and TIL, suggesting that circulating pTR-Ts may already exhibit features of a tissue-resident and activated phenotype.

(New Supplementary Figure 6, Averaged clone-level protein expression of TRTPBL markers (CD49a, CD49b, HLA-DR, and CD45RA) between pTR-Ts in PBMCs and those in TILs stratified by patient. Each dot and line represents a clonotype.)

11. The authors need to provide more information regarding the metastatic cohort related to number of cells that were sequenced from each patient and the UMPA distribution of clusters for each patient.

We agree. We have added information regarding the number of cells sequenced from each patient in revised Supplementary Table 9. We also provide the cluster distribution across patients in Supplementary Figure 7.

(Supplementary Table 9, Patient characteristics of the paired PBMC and TILs cohort, related to Figure 4.)

ID	Sex	Age at sample collection	Histology	Smoking status	Driver mutation	PD L1 TPS	Clinical stage	Treatment	Best objective response	Response	Progression-free survival, days	Cell number
G16	M	73	Adenocarcinoma	Smoker/Ex-smoker	No	50%	cT3N0M1c StageIVB	CBDCA +Pemetrexed+Pembrolizumab	Partial response	Responder	714	Pre: 1177 Post: 8651
G17	M	66	Adenocarcinoma	Smoker/Ex-smoker	No	<1%	cT1aN2M0 StageIIIA	CBDCA +Paclitaxel+Bevacizumab+Atezolizumab	Partial response	Responder	742	Pre: 2767 Post: 17050
G20	M	72	Adenocarcinoma	Smoker/Ex-smoker	No	1-49%	cT4N0M0 StageIIIB	CBDCA +Pemetrexed+Atezolizumab	Partial response	Responder	350	Pre: 12381 Post: 24389
G38	M	52	Adenocarcinoma	Smoker/Ex-smoker	No	1-49%	cT2bN1M1a StageIVA	CBDCA +Pemetrexed+Atezolizumab	Partial response	Responder	299	Pre: 10583 Post: 15621
G8	M	73	Adenocarcinoma	Smoker/Ex-smoker	No	1-49%	cT1cN0M1c StageIVB	CBDCA +Pemetrexed+Atezolizumab	Stable disease	Non-responder	147	Pre: 714 Post: 3843
G30	M	76	Adenocarcinoma	Smoker/Ex-smoker	No	<1%	cT2aN3M1a StageIVA	CBDCA +Pemetrexed+Atezolizumab	Stable disease	Non-responder	130	Pre: 9266 Post: 16358
G37	M	50	Adenocarcinoma	Smoker/Ex-smoker	No	1-49%	cT3N2M1c StageIVB	CBDCA +Paclitaxel+Bevacizumab+Atezolizumab	Progressive disease	Non-responder	42	Pre: 11779 Post: 19492
G58	M	71	Adenocarcinoma	Smoker/Ex-smoker	No	1-49%	cT1bN0M1a StageIVA	CBDCA +Abraxane+Atezolizumab	Stable disease	Non-responder	170	Pre: 5827 Post: 11475

(New Supplementary Figure 7, (A) Pre and post treatment single-cell CD8⁺ PBMCs from 8 patients who received immune-checkpoint therapy were projected onto the UMAP image (Figure 2A) (B) Distribution of total CD8⁺ T cells across clusters in each patient, shown separately by treatment phase (pre- and post-treatment) and response status)

12. The Results for Figure 4 claim to show genes that are significantly different between R and NR TR-T, however only 2 genes actually have an adjusted p-value <0.05. If the authors insist, these genes should also be confirmed at the patient level to ensure the differences in gene expression are not driven by patient-specific effects.

We appreciate the reviewer's constructive suggestion. We revised Figure 4C to show adjusted p-values instead of unadjusted ones. Notably, the number of DEGs increased to 6 following the refinement of pTR-T signature as described in our response to the comment #6. We further evaluated the consistency of these differentially expressed genes (DEGs) across individual patients, and found that some DEGs (GRHPR, MGMT, and AKR7A2) were consistently downregulated. We have revised the *Result* section accordingly, and added Supplementary Figure 8A and 8B.

(Page 22, Line 364-373 in the *Result* section of the revised manuscript)

Patient-level evaluation of these DEGs (Supplementary Figure 8A) revealed consistently downregulated genes in non-responders, including GRHPR (glyoxylate reductase/hydroxypyruvate reductase), MGMT (methylguanine DNA methyltransferase), and AKR7A2 (aldo-keto reductase). Although the roles of these genes in T cells remain largely unknown, their involvement in oxidative stress and DNA damage response suggests that metabolic adaptation of circulating TR-Ts may differ between responders and non-responders.

(Previous Figure 4C)

(Revised Figure 4C, Volcano plots showing DEGs between responders' and non-responders' pTR-Ts)

(New Supplementary Figure 8A, Patient-level expression of differentially expressed genes (Figure 4C) between responders' and non-responders' pTR-Ts in pre-treatment samples (Wilcoxon rank-sum test))

(New Supplementary Figure 8B, Patient-level expression of differentially expressed proteins (Figure 4D) between responders' and non-responders' pTR-Ts in pre-treatment samples (Wilcoxon rank-sum test))

13. The authors need to validate that the Results shown in Fig. 4G are consistent across patients

We completely agree with this point. To validate the findings previously shown in Figure

4G, we performed statistical analyses and presented the results in the revised Figure 4F, along with Supplementary Figures 8C and 8D, which illustrate patient-level distributions of pTR-Ts across clusters. We found that the transition from clusters 2 and 11 (12 in the previous manuscript) to clusters 3 and 8 (8 in the previous manuscript) was statistically significant in responders but not in non-responders. To clarify this point, we have revised the Results section, replaced Figure 4G with the new Figure 4F, and added Supplementary Figures 8C and 8D.

(Page 23, Line 384-386 in the *Result* section of the revised manuscript)

In responders, pTR-Ts in clusters 2 and 11 were significantly reduced after treatment, accompanied by a marked increase in clusters 3 and 8. In contrast, such shifts were not observed in non-responders (Figure 4F).

(Previous Figure 4G)

(New Figure 4F, Proportional changes in phenotypic clusters in pTR-Ts after ICI treatment in responders and non-responders. (paired t-test))

(New Supplementary Figure 8, (C) Patient-level pTR-T distribution shown separately by treatment phase (pre- and post-treatment) (D) Distribution of pTR-T cells across clusters in each patient, shown separately by treatment phase and response status. The proportion of pTR-T cells within each cluster is displayed for individual patients.)

Minor

1. There is no table provided showing the number of cells that were sequenced from each individual sample, making it difficult to assess the generalizability of the results

We agree. We have provided the number of cells sequenced from each individual sample

in Supplementary Tables 1 and 9.

(Supplementary Table 1)

Supplementary Table 1. Patient characteristics of paired PBMC and TILs cohort, related to Figure 1.

Patient ID	(Omit)	Number of cells (TIL)	Number of cells (PBMC)
TS4		4,212	6,895
TS5		2,243	3,368
TS7		5,390	200
TS13		7,181	2,396
TS16		6,335	8,323
TS17		271	6,890
TS19		289	3,483
TS20		1,818	8,342
TS25		1,770	6,675

(Supplementary Table 9 was shown in the response on the comment #11.)

2. *Most of the CMV-specific clones in the public databases are derived from CD4+ T cells. There can be quite a lot of overlap in Vbeta sequences between CD4 and CD8 T cells despite recognizing distinct antigens. Although the Methods indicate that “clones” were identified based on alpha and beta sequences, most public databases only contain beta sequences. The authors need to clarify this point.*

Thank you for your thoughtful comment. To filter out CD4-derived clones, we selected CDR3 sequences against MHC class I-restricted epitopes. Both α and β sequences were used for defining clonotypes in our dataset. On the other hand, for identifying viral-reactive T cells, a match in either the α or β chain to known viral CDR3 sequences was considered sufficient, as both chains are not always available in public databases. We apologize for any confusion. To clarify, we have revised the Methods section as follows:

(Page 38, line 639-643 in the *Method* section of the revised manuscript)

VR-Ts were identified by matching TCR sequences to entries in the VDJdb database (32). CDR3 sequences against MHC class I-restricted epitopes were selected. A match in either

the α or β chain to known viral CDR3 sequences (i.e., CMV, EBV, or influenza) was considered sufficient for annotation as VR-Ts, as both chains were not always available in the database.

3. *This study seems to contradict prior findings that flu-specific T cells in lung cancers are of a tissue-resident memory phenotype. This point is not addressed.*

Thank you for your insightful comment. As you noted, flu-specific T cells in our study exhibited either a tissue-resident memory or effector memory phenotype, which is consistent with previous reports. To clarify this point, we revised the corresponding figure to display viral-reactive T cells separately according to the type of virus.

(Previous Figure 1E)

(New Figure 1E, Distribution of VR-Ts on UMAP. VR-Ts were defined as clones containing complementary determining region 3 sequences reactive to influenza, CMV, or EBV)

4. *In Figure 2I, "Rami et al" should be "Yossef et al"*

Thank you for your correction. We have revised that point.

(Revised Figure 2I)

5. Lines 342-351 seemingly come out of nowhere. It is unclear how this relates to the data that were discussed in the prior sentences.

We apologize for the confusion. In this paragraph, we performed DEG and DEP analyses comparing pTR-Ts from responders and non-responders, aiming to characterize the phenotypic features of pre-treatment pTR-Ts. To clarify this point, we have reorganized the paragraph to simplify the results.

(Lines 342-351 in the previous manuscript)

For clinical utilization, we performed DEP analysis (Figure 4E and Supplementary Table 9). CD38, a cell surface glycoprotein involved in immune regulation and NAD⁺ metabolism, and CD101, immunoglobulin superfamily member 2 (*IGSF2*), were top ranked in non-responders. Consistent with this, the immunosuppressive roles of these molecules in resistance to PD-1 blockade therapies have been previously reported (37-41). *CD38* and *IGSF2* did not reach statistical significance in the transcript-level analysis (Supplementary Table 8), indicating the significance of protein-level analysis for biomarker detection. These results suggested that assessing the expression of these markers, specifically in pTR-Ts, could enhance the accuracy of ICI response prediction.

(Page 37-38, line 364-379 in the *Result* section of the revised manuscript)

Although the overall UMAP distribution was similar between responders and non-responders, we identified several DEGs and DEPs when comparing gene expression profiles between the two groups (Figure 4C, 4D, Supplementary Tables 10 and 11). Patient-level evaluation of these DEGs (Supplementary Figure 8A) revealed consistently downregulated genes in non-responders, including *GRHPR* (glyoxylate reductase/hydroxypyruvate reductase), *MGMT* (methylguanine DNA methyltransferase), and *AKR7A2* (aldo-keto reductase). Although the roles of these genes in T cells remain largely unknown, their involvement in oxidative stress and DNA damage response suggests that metabolic adaptation of circulating TR-Ts may differ between responders and non-responders. DEP analysis and patient-level analysis further revealed that HLA-DR and CD38 were consistently upregulated in non-responders (Supplementary Figure 8B). CD38, a cell surface glycoprotein involved in immune regulation and NAD⁺ metabolism, has been reported to contribute to resistance to PD-1 blockade therapies, consistent with the immunosuppressive roles of these molecules (39-43). Therefore, assessing these markers in pTR-Ts could be used for ICI response prediction. These phenotypic differences may further reveal underlying mechanisms that contribute to

clinical outcomes. Patient-level DEP analysis further revealed that HLA-DR and CD38 were consistently upregulated in non-responders (Supplementary Figure 8B). CD38, a cell surface glycoprotein involved in immune regulation and NAD⁺ metabolism, has been reported to contribute to resistance to PD-1 blockade therapies (39-43). Therefore, assessing these markers in pTR-Ts could be used for ICI response prediction. These phenotypic differences may further reveal underlying mechanisms that contribute to clinical outcomes.

6. In general, this study fails to report the findings within the context of prior work on tumor-specific TIL in lung cancer.

We thank the reviewer for this insightful comment. Indeed, several important studies have characterized tumor-specific TILs in lung cancer, highlighting their functional heterogeneity, exhaustion phenotypes, and prognostic relevance in the context of immune checkpoint blockade. These studies focused on tumor-specific TILs, but their circulating counterpart has not been evaluated. Here, we provided novel markers of circulating TR-Ts, and showed their transcriptional reprogramming following PD-1 blockade therapy. In light of this, we have revised the *Discussion* to better situate our findings within this existing body of work.

(Page 26, line 430-434 in the *Discussion* section of the previous manuscript)

While previous studies have focused on TILs, demonstrating that ICI response links to the phenotypic change and reinvigoration of TILs (4-7, 46, 47), our study provides novel evidence that circulating TR-Ts in responders undergo transcriptional reprogramming following PD-1 blockade therapy, shifting toward a previously uncharacterized population of stem-like effector memory T cells.

(Page 28, line 458-463 in the *Discussion* section of the revised manuscript)

Previous studies have focused primarily on TR-Ts in TILs, demonstrating that these cells express exhaustion and tissue-resident markers such as CD39, CD103 and CXCL13 (4-7, 51, 52). However, it remains unclear how these TR-Ts in TILs respond to PD-1 blockade. In this context, our study provides novel evidence that circulating TR-Ts in responders undergo transcriptional reprogramming following PD-1 blockade, shifting toward a previously uncharacterized population of stem-like effector memory T cells.

We believe that incorporating your advice into the R1 version has substantially improved

the manuscript. Thank you once again.

Reviewer #2 (Remarks to the Author):

The majority of T cells in patients with cancer are not specific to tumor antigens, so unravelling the biology of tumor specific T cell responses, as well as developing predictive tests and identifying sources of T cells for therapy, rely on the identification of tumor specific T cell subsets. In recent years, transcriptional signatures have been identified that differentiate tumor specific T cells in tumors from tumor non-specific bystander T cells, but less work has been done on the defining characteristics of these cells in the peripheral blood, where they are more accessible to analysis, possibly less functionally impaired, but significantly rare. This work uses single cell RNA sequencing to identify intratumoral CD8⁺ T cells with a transcriptional signature of tumor antigen reactivity in a cohort of smoking patients with non-small cell lung cancer, and then utilize the unique TCRVb gene rearrangements that mark specific T cell clones to interrogate the phenotype of these cells in the peripheral blood. They identify transcriptional and surface marker signatures enriched for these cells in the blood, find that these cells expand in the blood and change in phenotype on successful treatment with PD-1 inhibitors, and validate this subset contain cells specific to a known tumor antigen in a patient with melanoma. These include a phenotype of CD49a/b and HLA-DR that is novel to this study and may lend insight into the potential trafficking of these cells. They find that tumor specific CD8 T cells of known specificity to a model antigen in a mouse model of PD-1 inhibitor sensitive melanoma have similar characteristics. They identify pre-treatment lack of CD38 expression in this blood subset and expansion of clones in this subset as modest biomarkers of treatment response. While this is not the first description of similar cell populations, and the direct clinical consequences of using this cell population as a biomarker or source of therapeutic T cells is likely to be modest, this work is well done, a valuable contribution to the field, and the weaknesses identified are relatively minor and can be addressed by changes in emphasis in the results and discussion.

Thank you so much for your insightful and supportive comment to our study. We believe this study contribute to better understanding of the circulating tumor-reactive T cells and its dynamics in response to immunotherapy. In response to comment #1, #2, and #4, we have added a paragraph discussing the limitations of our study in the *Discussion* section. Below, we provide detailed, point-by-point responses to each of the reviewer's comments and indicate how we have addressed them in the revised manuscript.

Minor points:

1-pseudotime analysis does not demonstrate a differentiation pathway (in fact it assumes one), and the authors should be cautious about over-interpreting their findings in figure 3. It is possible, maybe even likely that most of the phenotypic difference between intratumoral tumor specific cells and their counterparts in the blood is exposure to antigen in the tumor, and it is not at all clear that peripheral T cells are a source of intratumoral T cells during treatment.

Thank you so much for your valuable comment. We completely agree with this point. We have revised the *Abstract*, *Result* and *Discussion* to clarify this limitation and to avoid over-interpretation.

(Page 5 (Abstract), line 75-77 in the previous manuscript)

Trajectory analysis revealed that the cTR-Ts exhibited a progenitor-like phenotype, suggesting their role as the origin of tumor-infiltrating TR-Ts.

(Page 5 (Abstract), line 75-77 in the revised manuscript)

Trajectory analysis revealed that the cTR-Ts exhibited a progenitor-like phenotype, suggesting a potential developmental relationship with tumor-infiltrating TR-Ts.

(Page 19, line 307-311 in the *Result* section of the previous manuscript)

Collectively, our trajectory analysis suggests circulating TR-Ts' role as the origin of tumor-infiltrating TR-Ts rather than exiting.

(Page 18-19, line 345-347 in the *Result* section of the revised manuscript)

Collectively, our trajectory analysis demonstrated phenotypic and transcriptional distinctions and similarity between circulating and intratumoral TR-Ts.

(Page 20, line 348-350 in the *Result* section of the previous manuscript)

We investigated whether circulating pTR-T represents T-cell egress from a tumor or is a precursor of tumoral pTR-T.

(Page 20, line 327-329 in the *Result* section of the revised manuscript)

We investigated whether circulating pTR-Ts resemble specific differentiation states of tumoral pTR-Ts.

(Page 21, line 342-347 in the *Result* section of the revised manuscript)

Notably, despite these variable changes in TRTPBL markers, TRM-associated and activation markers were consistently expressed at relatively high levels in both peripheral blood and TIL, suggesting that circulating pTR-Ts may already exhibit features of a tissue-resident and activated phenotype. Collectively, our trajectory analysis highlighted phenotypic and transcriptional similarities and differences between circulating and intratumoral TR-Ts.

(Page 26, line 436-438 in the *Discussion* section of the previous manuscript)

Our study strongly indicates that peripheral blood serves as a significant source of functional, proliferative T cells that mediate antitumor effects in response to PD-1 blockade therapy.

(Page 28, line 465-467 in the *Discussion* section of the revised manuscript)

Our study suggests that peripheral blood contains functional, long-lived T cells that may contribute to antitumor effects during PD-1 blockade therapy.

(Page 28-29, line 488-492 in the *Discussion* section of the revised manuscript)

It should be noted that trajectory analysis does not provide definitive evidence of a differentiation pathway, and it remains unclear whether the phenotypic differences between circulating and intratumoral TR-Ts arise from spatiotemporal lineage relationships or from exposure to tumor antigens and the microenvironment.

2-From their data, it appears the clinical utility of analysis of these cell populations for pre-treatment or early treatment prognostication will be limited, but that is unfortunately the case with most such biomarkers, and does not detract from what these correlations suggest about the biology.

We agree with this point. We acknowledge that the predictive value of variables related to circulating TR-Ts was modest and insufficient for clinical application. Nevertheless, we believe it is still important to report these findings, as they highlight the biological relevance of circulating TR-Ts in the context of ICI therapy. To clarify this point, we have addressed this issue in a newly added paragraph discussing the limitations of our study in the *Discussion* section, as follows.

(Page 33, line 553-559 in the *Discussion (Limitation)* section of the revised manuscript)

Second, the predictive performance of variables associated with TRTPBL markers was

modest and not sufficient for immediate clinical application. Given the limited number of patients, our single-cell analysis identified only a few potential biomarker candidates among ICI-treated individuals. Nonetheless, our study underscores the biological relevance of circulating TR-Ts to ICI response, suggesting that further investigations focusing on this minor PBMC compartment may help identify new predictive biomarkers.

3-While it is tantalizing that peripheral blood T cell populations could be used as a source of T cells for therapeutic products, it is not clear whether these cells will be sufficiently enriched in the relevant subsets or sufficiently common to make this happen.

We appreciate your insightful comment. We agree that it remains uncertain whether sufficient numbers of TR-Ts can be obtained from PBMCs for therapeutic applications. To demonstrate that TR-Ts can be enriched, we performed additional experiments in which PBMCs were stained with TRTPBL markers, sorted by flow cytometry, and subjected to single-cell TCR sequencing. These analyses confirmed that TR-Ts were markedly enriched within the sorted population (Figure 2I). Nevertheless, whether the number of TR-Ts obtained through this approach is sufficient for therapeutic purposes requires further investigation. We have added Figure 2I and this discussed this issue in the *Discussion* section of the revised manuscript as follows.

(Page 17-18, line 286-291 in the *Result* section of the revised manuscript)

Finally, we sorted TRTPBL marker-positive cells and performed single-cell TCR sequencing. The sorted population showed a significant enrichment of pTR-Ts at both the clonal and patient levels (Figure 2I). The concentration of pTR-Ts increased 2.7-13.5 folds. Notably, sorting based on TRTPBL markers successfully identified 15 additional pTR-T clones that were not detected in total CD8⁺ PBMCs scRNA dataset, as well as pTR-Ts in one patient in whom these cells had not previously been detected.

(Page 31, line 508-513 in *Discussion* (Limitation) section of the revised manuscript)

Notably, circulating TR-Ts remain a relatively small population even after TRTPBL marker-based enrichment. Further optimization of enrichment and expansion protocols will be essential to achieve sufficient yield for clinical application. Nevertheless, our findings highlight that peripheral blood can serve as a feasible and antigen-agnostic source of TR-Ts with translational potential.

(Page 35, line 587-590 in *Method* section of the revised manuscript)

For TRTPBL marker-enrichment experiment, anti-CD45RA vF450 (HI100, TONBO Biosciences), anti-CD49b FITC (P1E6-C5, BioLegend), anti-CD49a (TS2/7 BioLegend), and anti-HLA-DR PE/dazzle 594 (L243, BioLegend) were added to the panel to enrich TRTPBL marker⁺ cells.

(Figure 2I, Flow-cytometry enrichment of pTR-Ts based on TRTPBL markers at clone-level (left) and patient-level (right). (paired *t*-test))

4-The relative lack of validation of tumor antigen specificity of the identified putative tumor specific clonotypes is a weakness, but as the authors point out, such studies are challenging and essentially impossible to do comprehensively, and the work they present in melanoma as validation is adequate for the purposes of this paper.

Thank you for your supportive feedback. We agree that this is an important limitation of this study. We chose to use gene signature-based TR-T prediction rather than experimental affinity test as it is impossible to do comprehensively. To clarify this point, we have added the following text to the Discussion.

(Page 33, line 546-553 in the *Discussion* (Limitation) section of the revised manuscript)
This study has several limitations. First, we did not experimentally validate the tumor antigen specificity of pTR-Ts. Comprehensive functional testing of TCR antigen specificity is technically challenging and practically infeasible for large numbers of clones. Instead, we utilized a gene signature-based TR-T prediction approach, which has demonstrated excellent predictive performance in prior studies. External validation in a melanoma dataset further supports the robustness of our findings. Future studies will be

needed to test the generalizability of these markers across different cancer stages and tumor types.

We believe that incorporating your advice into the R1 version has substantially improved the manuscript. Thank you once again.